# Visually Guided Generative Text-Layout Pre-training
# for Document Intelligence

**Zhiming Mao**[1,2], **Haoli Bai**[3], **Lu Hou**[3],

Jiansheng Wei[3], Xin Jiang[3], Kam-Fai Wong[1,2], Qun Liu[3]

[1]The Chinese University of Hong Kong, Hong Kong, China

[2]MoE Key Laboratory of High Confidence Software Technologies, China

[3]Noah's Ark Lab, Huawei Technologies

[1,2]{zmmao,kfwong}@se.cuhk.edu.hk

[3]{baihaoli, houlu3, weijiansheng, Jiang.Xin, qun.liu}@huawei.com

## Abstract

Prior study shows that pre-training techniques can boost the performance of visual document understanding (VDU), which typically requires models to gain abilities to perceive and reason both document texts and layouts (e.g., locations of texts and table-cells). To this end, we propose visually guided generative text-layout pre-training, named ViTLP. Given a document image, the model optimizes hierarchical language and layout modeling objectives to generate the interleaved text and layout sequence. In addition, to address the limitation of processing long documents by Transformers, we introduce a straightforward yet effective multi-segment generative pre-training scheme, facilitating ViTLP to process word-intensive documents of any length. ViTLP can function as a native OCR model to localize and recognize texts of document images. Besides, ViTLP can be effectively applied to various downstream VDU tasks. Extensive experiments show that ViTLP achieves superior performance over existing baselines on benchmark VDU tasks, including information extraction, document classification, and document question answering.[1]

## 1 Introduction

Processing and reasoning document images with dense texts (e.g., scanned PDF files, digital forms, and spreadsheets) is a persistent yet challenging task for the research community and industry (Katti et al., 2018; Majumder et al., 2020; Li et al., 2021a). Advances in multimodal pre-training substantially improve the performance of visual document understanding (VDU) (Xu et al., 2020, 2021; Gu et al., 2021; Appalaraju et al., 2021; Wang et al., 2022a). These pre-training methods typically take multimodal inputs of given document images: i) visual features, ii) document OCR texts, and iii) spatial layouts of document elements, e.g., 2D coordinates of texts and table-cells. Among these elements,

---

[1]Code and checkpoints will be released once published.

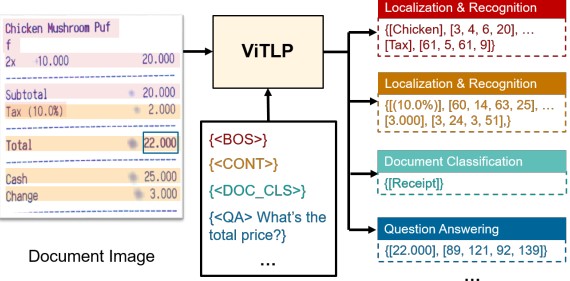

Figure 1: An overview workflow of the proposed ViTLP. Given a document image as input, ViTLP can generate sequences of text and layout (i.e., word bounding boxes) for various VDU tasks with task-specific prefixes.

spatial layout information plays an essential role in connecting visual and textual features, as well as developing a thorough understanding of document structures (Lee et al., 2022; Wang et al., 2022b).

Though effective, the performance of most existing VDU approaches relies heavily on the OCR pipelines, because the pre-processed OCR texts and corresponding 2D coordinates are used as intermediate inputs to pre-trained VDU models. The external OCR pipelines may produce incorrect or incomplete recognition results, which cannot be jointly optimized by the gradient back from VDU models. Another research line (Kim et al., 2022; Lee et al., 2023b) explores pre-training VDU models solely based on image inputs. Despite no OCR errors introduced, these methods focus on understanding texts from raw document images but dismiss layout information modeling. Since the spatial information contained in layout locations is not exploited, it may hinder the models from understanding complex document structures, especially for documents containing nested paragraphs, forms, and tables.

In this work, we propose **Vi**sually guided generative **T**ext-**L**ayout **P**re-training (ViTLP) to jointly model text and layout information from document images. As shown in Figure 1, ViTLP can localize, recognize, and understand visual document texts

given the input document image and task prefixes. To achieve this goal, ViTLP is pre-trained to generate *unified text-layout sequences* from document images. Since natively generating text and layout tokens in a flattened sequence is *token-inefficient* (see Sec. 2.1), we introduce hierarchical generation modules to achieve both effective and efficient text-layout sequence generation. To the best of our knowledge, ViTLP is the first attempt to learn OCR (i.e., text localization and recognition) and VDU (i.e., document understanding) abilities in a unified generative text-layout pre-training framework.

Besides, ViTLP is designed to handle long documents with intensive texts. Long document processing is ubiquitous in real-world scenarios. However, existing pre-trained models are constrained to certain token limits of input sequences. For instance, LayoutLMv2 (Xu et al., 2021) accepts the maximum inputs of $512$ word tokens using a BERT-structure encoder. In both pre-training and fine-tuning, the exceeded text tokens are truncated, leading to incomplete document information modeling. To tackle this issue, we introduce a **multi-segment pre-training scheme** which divides the target text-layout sequence into consecutive segments to perform generative pre-training. Given that the full document information is already encoded in visual representations, ViTLP employs the suffix tokens from previous segments as prefix prompts to generate next-segment tokens. This multi-segment pre-training scheme further enables ViTLP to process documents of arbitrary length in fine-tuning. Notably, our multi-segment generation scheme retains the intact transformer architecture. Thus, it is more feasible than other long-document modeling workarounds, e.g., *sparse attention* (Beltagy et al., 2020) and *memory modules* (Bulatov et al., 2022), which need to modify the Transformer architecture and may affect the capacity of pre-trained models.

We evaluate the performance of ViTLP on a variety of OCR and VDU tasks. Empirical results demonstrate that ViTLP can generally achieve superior performance on both OCR and VDU tasks. For instance, ViTLP achieves the 95.16 F1 score on CORD information extraction and $95.28\%$ accuracy on RVL-CDIP document classification, both of which surpass existing approaches. Meanwhile, as ViTLP can provide regions of interest (ROI) for the generated text, this also helps in some VDU tasks (e.g., document question answering) to be more explainable and reliable.

## 2 Approach

### 2.1 Problem Formulation

We study multimodal pre-training for visual document modeling. As widely studied (Xu et al., 2020, 2021; Appalaraju et al., 2021; Li et al., 2021b; Powalski et al., 2021; Wang et al., 2022a; Huang et al., 2022; Wang et al., 2022b), document images $\mathbf{V}$, texts $\mathbf{T}$, and layouts $\mathbf{L}$ are three fundamental modalities for visual document modeling.

**Unified Text-Layout Generation** We cast the pre-training objective on visual documents as text-layout sequence (i.e., $\{\mathbf{T}; \mathbf{L}\}$) generation conditioned on document images $\mathbf{V}$. The document texts $\mathbf{T}$ are represented as word-token sequences. The layouts $\mathbf{L}$, following prior studies (Xu et al., 2020, 2021), can be represented by *location bounding boxes* of words. Instead of generating two separate sequences of $\mathbf{T}$ and $\mathbf{L}$, ViTLP generates the texts with corresponding layout locations in a sequence of interleaved text-layout tokens, which facilitates compact multimodal interaction between texts and layouts. For the $i$-th word of a document, its text-layout tokens $\{\mathbf{T}; \mathbf{L}\}_i$ are represented as

$$\{\mathbf{T}; \mathbf{L}\}_i = \big\{\{\boldsymbol{w}\}_i, \{z_{x1}, z_{y1}, z_{x2}, z_{y2}\}_i\big\}, \quad (1)$$

where $\{\boldsymbol{w}\}_i$ denotes the BPE tokens (Radford et al., 2018) of the $i$-th word, $\{z_{x1}, z_{y1}, z_{x2}, z_{y2}\}_i \in \mathbb{Z}_+^4$ are the corresponding left-top and right-bottom bounding box coordinates. Given a document with $N$ words, the objective is to maximize the likelihood function $\log p(\mathbf{T}; \mathbf{L}|\mathbf{V})$ which can be decomposed as autoregressive text and layout modeling:

$$\log p(\mathbf{T}; \mathbf{L}|\mathbf{V}) = \sum_{i=1}^{N} \underbrace{\log p(\mathbf{T}_i|\mathbf{T}_{<i}, \mathbf{L}_{<i}, \mathbf{V})}_{\text{Text-modeling Term}}$$
$$+ \underbrace{\log p(\mathbf{L}_i|\mathbf{T}_{\leq i}, \mathbf{L}_{<i}, \mathbf{V})}_{\text{Layout-modeling Term}}. \quad (2)$$

Note that Eq. (2) shares similar ideas with Chen et al. (2021), where word and bounding box generation can be formulated as language modeling on a unified text-layout sequence. However, it is in fact nontrivial to generate sequences as in Eq. (1), because real-world documents commonly contain intensive texts, generating each word followed by four coordinate tokens in a long flattened sequence is especially **token-inefficient**. This would bring prohibitive computational and space overhead[2] to the Transformer-based text-layout decoder.

---

[2] Recall that both the computational and space complexities of Transformers are quadratic $\mathcal{O}(L^2)$ in sequence length $L$.

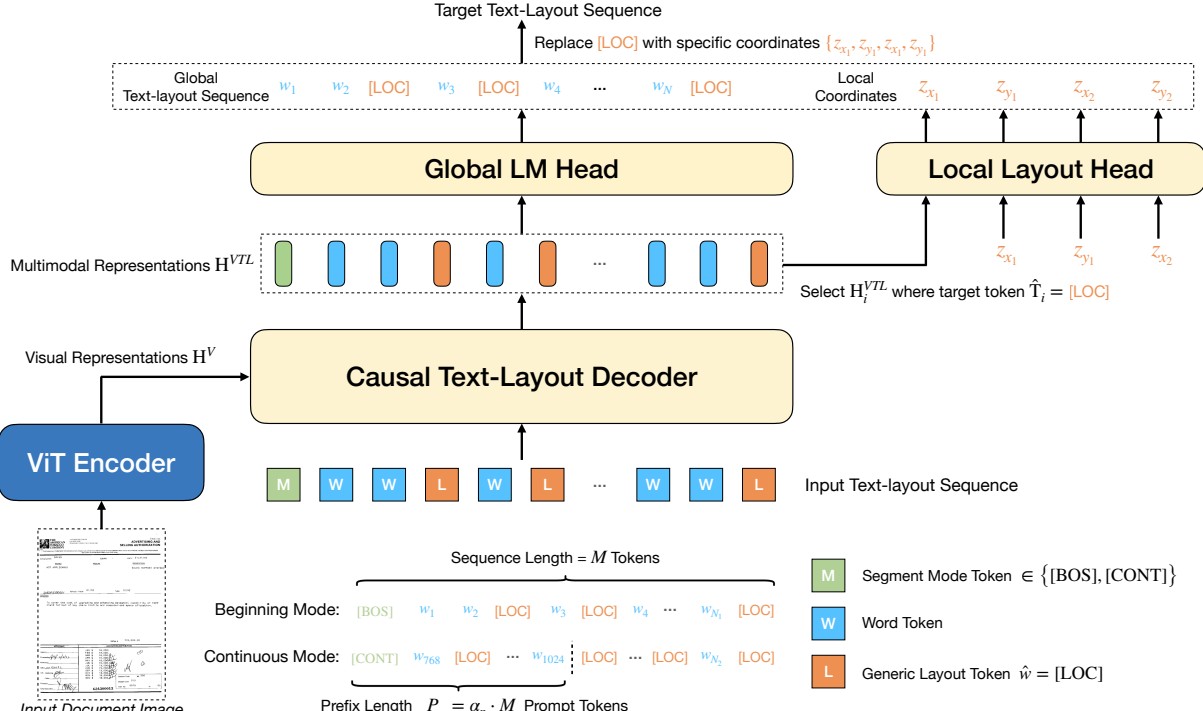

Figure 2: Overview of the ViTLP architecture. ViTLP is a generative pre-training model that performs autoregressive text-layout modeling conditioned on visual document inputs. ViTLP adopts hierarchical decoder heads to generate target text-layout sequences in a *global-to-local* manner. The segment mode tokens $\in \{[\text{BOS}], [\text{CONT}]\}$ prompt the beginning and continuous modes of generation, respectively.

## 2.2 Model Architecture

The architecture of ViTLP is shown in Figure 2. ViTLP employs an encoder-decoder framework to encode document images $\mathbf{V}$ and generate target text-layout sequences $\{\mathbf{T}; \mathbf{L}\}$. Specifically, given an input document image $\mathbf{V}$, ViTLP employs a vision transformer (ViT) (Dosovitskiy et al., 2020) to learn visual representations $\mathbf{H}^V \in \mathbb{R}^{|V| \times d}$, where $|V|$ is the ViT patch number and $d$ is the hidden size. The decoder receives the visual representations $\mathbf{H}^V$ and generates the unified text-layout sequence $\{\mathbf{T}; \mathbf{L}\}$. To address the *token-inefficiency* issue discussed in Sec. 2.1, we design the *global-to-local* text-layout generation process as follows.

### 2.2.1 Global Text-Layout Modeling

Instead of directly generating the text-layout sequence as in Eq. (1), we first replace the bounding box coordinates $\{z_{x1}, z_{y1}, z_{x2}, z_{y2}\}$ with a generic layout location token $\hat{w} = [\text{LOC}]$. This integrates the mixed text-layout sequence $\{\mathbf{T}; \mathbf{L}\}$ to unified language modeling. Given the original vocabulary $\mathcal{V}$, the **global text-layout sequence** $\hat{\mathbf{T}}$ derives from the augmented vocabulary $\hat{\mathcal{V}} = \mathcal{V} \cup [\text{LOC}]$. The layout token embeddings $\mathrm{E}_{[\text{LOC}]}$ are computed as

$$\mathrm{E}_{[\text{LOC}]} = \big[\mathrm{E}_x(z_{x1}), \mathrm{E}_y(z_{y1}), \mathrm{E}_x(z_{x2}), \mathrm{E}_y(z_{y2})\big],$$

where $\mathrm{E}_x(\cdot) \in \mathbb{R}^{\frac{d}{4}}$ and $\mathrm{E}_y(\cdot) \in \mathbb{R}^{\frac{d}{4}}$ denote the x- and y-axis spatial embeddings. Besides, the word tokens are embedded by $\mathrm{E}_w(\cdot) \in \mathbb{R}^d$. Given a document of $N$ words and the corresponding bounding boxes, the text-layout input embeddings are represented as $\mathbf{H}^{TL} = \{\mathrm{E}_w, \mathrm{E}_{[\text{LOC}]}\} \in \mathbb{R}^{|\hat{\mathbf{T}}| \times d}$.

The ViTLP text-layout decoder performs multimodal interaction among *visual*, *textual*, and *layout* information via the Transformer cross-attention

$$\mathbf{H}^{VTL} = \text{Transformer-Decoder}(\mathbf{H}^V, \mathbf{H}^{TL}).$$

For the $i$-th target token $\hat{\mathbf{T}}_i$, the multimodal decoder output $\mathbf{H}_i^{VTL}$ is fed to a linear language modeling (LM) head with the softmax function to compute the conditional generative probability

$$p(\hat{\mathbf{T}}_i | \hat{\mathbf{T}}_{<i}, \mathbf{V}) = \text{Softmax}\big(\text{Linear}(\mathbf{H}_i^{VTL})\big).$$

With the generic layout token $[\text{LOC}]$ incorporated, the text-modeling term in Eq. (2) is expressed as

$$\mathcal{L}_{\text{global-text}} = -\frac{1}{|\hat{\mathbf{T}}|} \sum_{i=1}^{|\hat{\mathbf{T}}|} \log p(\hat{\mathbf{T}}_i | \hat{\mathbf{T}}_{<i}, \mathbf{V}). \quad (3)$$

### 2.2.2 Local Layout Modeling

Local layout modeling aims to generate specific layout locations for each generic layout token $[\text{LOC}]$.

To capture the spatial relation among coordinates, we employ a simple sequential MLP layout head[3] to decode the short sequence of four layout coordinate tokens from the last hidden state of [LOC]. For notation simplicity, we denote $\{\mathbf{L}_{i,j}\}_{j=1}^4 = \{z_{x1}, z_{y1}, z_{x2}, z_{y2}\}_i$ as the corresponding layout coordinates of the [LOC] token at the $i$-th position, and its generative probability is modeled as

$$p(\mathbf{L}_{i,j}|\mathbf{L}_{i,<j}, \hat{\mathbf{T}}_{\leq i}, \mathbf{V}) = \text{Softmax}\big(\text{MLP}(\mathbf{H}_{i,<j})\big),$$

where $\mathbf{H}_{i,0} = \mathbf{H}_i^{VTL}$ is selected from the learned multimodal representations where $\hat{\mathbf{T}}_i = $ [LOC]. Here, we denote the index set of [LOC] tokens as $\mathcal{S}_L = \big\{i : \hat{\mathbf{T}}_i = [\text{LOC}] \,|\, i = 1, 2, ..., |\hat{\mathbf{T}}|\big\}$. The layout-modeling term in Eq. (2) is expressed as

$$\mathcal{L}_{\text{local-layout}} = -\log p(\mathbf{L}_i|\hat{\mathbf{T}}_{\leq i}, \mathbf{L}_{<i}, \mathbf{V}) \qquad (4)$$

$$= -\frac{1}{4|\mathcal{S}_L|} \sum_{i \in \mathcal{S}_L} \sum_{j=1}^4 \log p(\mathbf{L}_{i,j}|\mathbf{L}_{i,<j}, \hat{\mathbf{T}}_{\leq i}, \mathbf{V}).$$

In summary, with the global and local text-layout modeling in a hierarchy, the original pre-training objective in Eq. (2) evolves to

$$\mathcal{L} = \mathcal{L}_{\text{global-text}} + \mathcal{L}_{\text{local-layout}}. \qquad (5)$$

The *global-to-local* generation process aims to be effective and efficient for text-layout modeling. On effectiveness, the interleaved text-layout sequence modeling enables compact interaction between text and layout inputs, which can effectively fuse the information of text and layout modalities. On efficiency, suppose that the average BPE tokens of a document word are $|w|$, and the *compression ratio* of the text-layout sequence is $\frac{|w|+1}{|w|+4}$, i.e., four coordinate tokens are compressed to one. In our experiment datasets, the *compression ratio* is $0.48$.

## 2.3 Multi-segment Pre-training Scheme

Documents are usually intensive in text and layout, and it would be computationally intractable to fit the entire sequence into ViTLP. To process documents with arbitrary length, we propose a multi-segment pre-training scheme that divides the long sequence into multiple segments for generation. Since a document image already contains all necessary information of the text and layout, long document modeling is feasible based on the *visual representations* and *generation history context*.

Given the maximum sequence length of the decoder as $M$, we first divide the text-layout sequence

---

[3]Details of the layout head are in Appendix B.

into $K$ segmented sequences $\{\mathbf{S}_i\}_{i=1}^K$. The beginning segment $\mathbf{S}_1$ contains $M$ tokens to be generated, and the continuous segment $\mathbf{S}_{i>1}$ contains $\alpha_p \cdot M$ prefix tokens and $(1 - \alpha_p) \cdot M$ tokens to be generated. Here, $\alpha_p$ is the pre-defined prefix ratio. The overall generation process comprises beginning and continuous modes.

**Beginning Generation Mode.** In this mode, we prepend a special mode token [BOS] to the beginning sequence $\mathbf{S}_1$. The model then follows the objective in Eq. (5) to generate the first $M$ tokens.

**Continuous Generation Mode.** For the continuous segments $\mathbf{S}_{i>1}$, we prepend a special mode token [CONT] to the input sequence. $|P| = \alpha_p \cdot M$ prefix tokens are prepended to the input sequence. The $|P|$ **prefix tokens** of segmented sequence $\mathbf{S}_i$ come from the $|P|$ **suffix tokens** of the previous segmented sequence $\mathbf{S}_{i-1}$. These prefix tokens serve as the prompt of *generation history context* guiding the decoder to generate subsequent tokens from arbitrary locations of a document. The special token [EOS] is appended to the last segmented sequence $\mathbf{S}_K$ to signal the end of generation.

**Segmentation in Pre-training and Fine-tuning.** In pre-training, the segmented sequences of a long document are randomly scattered into different data batches. In this way, ViTLP learns to model the complete textual and layout information of a document, conditioned on different prefix history-token contexts. As for fine-tuning (and inference), ViTLP can also apply the multi-segment scheme to process long input/output text sequences, which is consistent with the pre-training phase. For example, OCR and information extraction on long document texts can be processed segment by segment.

## 2.4 Applications of ViTLP

### 2.4.1 OCR Text Localization and Recognition

Text localization and recognition (Li et al., 2023) are two fundamental functions of OCR engines. As ViTLP is pre-trained to generate texts and layouts (i.e., 2D bounding boxes) sequences from document images, it naturally supports text localization and recognition by means of zero-shot or fine-tuning on downstream OCR datasets. Thus ViTLP can be applied as an alternative to OCR engines.

### 2.4.2 Downstream VDU Tasks

**Information Extraction.** The information extraction task is formulated as sequential token-

labeling on the target texts given document images. Following BART (Lewis et al., 2020), we feed the decoder's final hidden states of a target word (with the layout token [LOC]) to a linear classifier which outputs the token-level classification label.

**Document Classification.** Given an input document image to the encoder, we feed a special prefix token [DOC_CLS] as input to the decoder, and output the corresponding label token.

**Document Question Answering.** Different from discriminative VDU models that perform extractive QA on pre-processed OCR results, ViTLP directly generates answers with the referring 2D layout locations in images. In this way, ViTLP can also provide explainable **grounding region of interest (ROI)** to justify the generated answers.

## 3 Experiments

### 3.1 Experiment Configurations

**Implementation Details** We implement ViTLP with a 12-layer ViT (Dosovitskiy et al., 2020) image encoder and a 6-layer text-layout decoder. The Transformer hidden size is $d = 768$ with 12 attention heads. In both pre-training and fine-tuning phases, the input image resolution is $1600 \times 1280$ with $32 \times 32$ ViT patch size, and the decoder segmented sequence length is $M = 1024$. Following LayoutLMv2 (Xu et al., 2021), the layout location coordinates are normalized into discrete bins of $[0, 1000]$, resulting that the vocabulary size of RNN layout-head is 1001. The multi-segment prefix ratio is set as $\alpha_p = 0.25$. We use the AdamW optimizer (Loshchilov and Hutter, 2019) to train ViTLP in 200K steps, with the batch size of 384 and initial learning rate of $2e\text{-}4$.

**Pre-training Data** Following previous work (Xu et al., 2021), we use IIT-CDIP Test Collection 1.0 (Lewis et al., 2006) containing 11M document images for pre-training. Five supplement datasets with 2.5M document images are also added to augment the diversity of pre-training data, including IAM (Marti and Bunke, 2002), SciTSR (Chi et al., 2019), PubLayNet (Zhong et al., 2019), DocBank (Li et al., 2020), and SynthDog (Kim et al., 2022). We use our internal OCR tool[4] to extract words with location coordinates from the IIT-CDIP and PubLayNet images. Words with locations are provided in IAM, SciTSR, and DocBank. Following

Kim et al. (2022), we generate 2M synthetic document images with text and layout annotations. Detailed data statistics are provided in Appendix A.1.

**Evaluation Tasks** We evaluate ViTLP on two types of document processing tasks: 1) *perception tasks* of document OCR and 2) *cognition tasks* of visual document understanding (VDU).

For OCR evaluation, we conduct two benchmark OCR sub-tasks, i.e., document text *localization* and *recognition*. We evaluate model performance on SROIE competition[5] Task #1 for text localization and Task #2 for text recognition. The text localization task is evaluated by DetEval protocol (Wolf and Jolion, 2006) which calculates the precision, recall, and F1 based on the *area of overlapping regions* between model predictions and ground-truth text coordinates. Given the model output and ground-truth words, the text recognition task evaluates the word-level precision, recall, and F1 based on exact word matches.

For VDU evaluation, we conduct three document understanding tasks. 1) *Form Understanding*. Given a document image and its word entities, it is a sequential labeling task to predict the BIO tags for each entity. We use FUNSD (Jaume et al., 2019) which contains 199 scanned forms, and the entities are labeled in four categories: *Header*, *Question*, *Answer*, and *Other*. FUNSD is divided into 149 images for training and 50 for testing. We report entity-level F1 as the evaluation score. 2) *Receipt Understanding*. We use CORD (Park et al., 2019) containing 800 training and 100 testing images of real-world receipts. The receipt entities are labeled in 30 categories. We use entity-level F1 for evaluation. 3) *Document Classification*. We conduct experiments on the RVL-CDIP dataset (Harley et al., 2015) containing 400K scanned documents in 16 classes. We adopt classification accuracy as the evaluation metric. For the sequence labeling tasks on FUNSD and CORD, we perform multi-segment fine-tuning on those samples whose entity-word sequences exceed the maximum decoder sequence length. This differs from previous work that truncates the input sequences into certain tokens, e.g., 512 tokens in LayoutLM (Xu et al., 2020).

Furthermore, we perform generative question answering on the DocVQA dataset (Mathew et al., 2021). DocVQA consists of over 12K document images with 50K question-answer pairs. Since the answer word locations are not provided in the train-

---

[4]https://www.huaweicloud.com/intl/en-us/product/ocr

[5]https://rrc.cvc.uab.es/?ch=13&com=tasks

ing data, we use an OCR tool to obtain the layout coordinates of document texts and then derive the answer word locations with heuristic text matching. In this way, we can feed the answers with layout coordinates to ViTLP for DocVQA fine-tuning.

## 3.2 OCR Evaluation Results

We compare ViTLP with representative OCR baselines on the SROIE 2019 benchmark (Huang et al., 2019). The text localization baselines include CRAFT (Baek et al., 2019), YOLO-v3 (Redmon and Farhadi, 2018), and CTPN (Tian et al., 2016). The text recognition baselines include BiLSTM-CTC (Graves et al., 2006), BiLSTM-ResNet, CTPN-CRNN (Shi et al., 2015), UNet-CRNN (Ronneberger et al., 2015), and TrOCR (Li et al., 2023). Unlike the conventional OCR models that first perform text localization and then use the localized text-regions to perform text recognition, ViTLP unifies text localization and recognition in a Transformer decoder and does not need ground truth text-region inputs in the recognition task.

Table 1 shows the OCR evaluation performance. ViTLP outperforms most baseline methods on both localization and recognition tasks. ViTLP underperforms TrOCR, given that TrOCR is a strong pre-trained model for two-stage OCR text recognition, while ViTLP performs text localization and recognition in one stage. Note that the SROIE training samples are few, i.e., only 626 images, and the provided input text coordinates are at textline-level, which are different from our word-level pre-training input format and thus render it challenging to fine-tune our model. Nonetheless, ViTLP can still achieve comparable performance by fine-tuning on the limited samples without extra data augmentation (Li et al., 2023) and adapting to output the textline coordinates that have never met in the pre-training phase. We also provide demonstrative zero-shot OCR samples in Appendix C.

## 3.3 VDU Evaluation Results

We compare ViTLP with competitive pre-training baselines including i) general pre-trained language model, RoBERTa (Liu et al., 2019), ii) disriminative VDU models: LayoutLM (Xu et al., 2020), SelfDoc (Li et al., 2021b), TILT (Powalski et al., 2021), LayoutLMv2 (Xu et al., 2021), and iii) generative VDU model, DONUT (Kim et al., 2022). Table 2 shows the VDU evaluation performance.

| Method | *Text Localization* | | |
| | Area-Precision | Area-Recall | Area-F1 |
|---|---|---|---|
| CRAFT | 62.73 | 59.94 | 61.31 |
| YOLO-v3 | 77.29 | 79.32 | 78.29 |
| CTPN | 81.14 | 87.23 | 84.07 |
| ViTLP | 90.12 | 90.77 | 90.45 |
| **Method** | ***Text Recognition*** | | |
| | **Word-Precision** | **Word-Recall** | **Word-F1** |
| CTPN-CRNN | 35.75 | 63.89 | 45.85 |
| BiLSTM-ResNet | 74.05 | 77.81 | 75.88 |
| BiLSTM-CTC | 83.38 | 87.37 | 85.33 |
| UNet-CRNN | 85.77 | 86.48 | 86.12 |
| TrOCR[†] | 95.89 | 95.74 | 95.82 |
| ViTLP | 90.61 | 90.72 | 90.66 |

Table 1: OCR text localization and recognition results on SORIE 2019. [†]TrOCR uses the ground-truth cropped image regions as inputs, whereas ViTLP performs text localization and recognition in one stage. All scores are reported in percentage.

**Information Extraction** In terms of the entity-level F1 scores on FUNSD, our model surpasses all baseline models by a clear margin, e.g., $+3.91$ points over LayoutLMv2. This result indicates that ViTLP can develop a thorough understanding of form structures in document images. In terms of CORD, ViTLP also achieves the highest F1 score. Note that ViTLP significantly outperforms DONUT, proving that explicit layout modeling is also necessary like language modeling for generative VDU models. Concretely, for the receipt images of CORD, entities with the same semantic label `<menu.price>` are always located in the same rightmost column of the receipt, sharing closed horizontal coordinates. Modeling the layout information could help generative VDU models better understand the document structures.

**Document Classification** As shown in Table 2, ViTLP achieves competitive performance on classification accuracy. We find that the performance of ViTLP, TILT, LayoutLMv2, and DONUT are nearly matched. This may be because document classification is a coarse-grained task, where the vision modality plays the most important role compared to texts and layouts. Regarding similar classification accuracy, OCR-free models, i.e., ViTLP and DONUT, are more flexible because no pre-processed OCR results are needed.

## 3.4 Further Discussions

### 3.4.1 Ablation Study

We conduct a comprehensive ablation study to investigate the effect of hierarchical text-layout mod-

| Pre-trained Model | # Param. | Maximum Doc-Length | FUNSD (F1↑) | CORD (F1↑) | RVL-CDIP (Acc↑) |
|---|---|---|---|---|---|
| RoBERTa$_{base}$ (Liu et al., 2019) | 125M | 512 | 66.48 | 93.54 | 90.06 |
| LayoutLM$_{base}$ (Xu et al., 2020) | 160M | 512 | 79.27 | – | 94.42 |
| SelfDoc (Li et al., 2021b) | 137M | 1024 | 83.36 | – | 93.81 |
| TILT$_{base}$ (Powalski et al., 2021) | 230M | 512 | – | 95.11 | 95.25 |
| LayoutLMv2$_{base}$ (Xu et al., 2021) | 200M | 512 | 82.76 | 94.95 | 95.25 |
| DONUT (Kim et al., 2022) | 259M | 1536 | – | 84.10 | 95.30 |
| ViTLP | 253M | unlimited | **86.67** | **95.16** | **95.28** |

Table 2: Multiple VDU evaluation tasks on form understanding (FUNSD), receipt understanding (CORD), and document classification (RVL-CDIP). "Maximum Doc-Length" stands for the maximum input tokens of a document that the pre-trained models can process.

| Ablation Model | FUNSD (F1) | CORD (F1) |
|---|---|---|
| w/o layout modeling | 81.33 | 91.73 |
| w/o multi-segment training | 85.86 | 94.96 |
| w/o hierarchical modeling | 85.58 | 95.04 |
| ViTLP | **86.87** | **95.16** |

Table 3: Ablation variant performance on the information extraction tasks.

| Generative Pre-trained Model | ANLS ↑ |
|---|---|
| DONUT (Kim et al., 2022) | 67.5 |
| DONUT† (official code + our OCR data) | 62.2 |
| ViTLP | 61.3 |

Table 4: Average Normalized Levenshtein Similarity (ANLS) between the generated answers and ground-truth answers. † denotes our re-implementation with the official codebase and our pre-training data.

eling and multi-segment pre-training scheme. We compare ViTLP with three variants: i) pre-training with the language modeling objective only, without the layout modeling objective; ii) truncating the long input sequences, without the multi-segment pre-training scheme; iii) generating four location tokens for each word in a flatten long sequence, without hierarchical text-layout modeling.

Table 3 displays the ablation performance. We observe that removing layout modeling leads to a substantial performance drop, i.e., 5.54 and 3.37 F1 drops on FUNSD and CORD. The results suggest that generative pre-training on the layout modality can enhance the document understanding ability of VDU models. Besides, truncating long document inputs without the multi-segment pre-training strategy leads to suboptimal performance. We believe that the multi-segment pre-training scheme enables ViTLP to model complete text and layout tokens of the pre-training corpora, which benefits pre-trained model performance. We can see that hierarchical text-layout modeling also brings incremental performance gain, implying its effectiveness in text-layout information fusion.

### 3.4.2 Generative Document VQA

**Performance and Analysis** Table 4 shows the DocVQA performance. ViTLP underperforms

DONUT by 6.2 ANLS scores. We analyze that the performance gap probably dues to two reasons: (1) As compared in Davis et al. (2022), pre-training data quality would significantly affect the downstream DocVQA task performance. This is because, different from discriminative models that perform extractive QA upon given OCR of document spans, generative models themselves need to develop strong language modeling abilities to organize accurate output answer words, which requires clean and diverse data for LM pre-training. We have benchmarked the official DONUT codebase with our pre-training data. The ANLS score frustratingly decreases from 67.5 to 62.2, indicating the pre-training data quality needs to be improved. (2) On the other hand, the DocVQA questions may not appear in document images. To keep consistent with the pre-training sequence format, we append the dummy coordinates $\{0, 0, 0, 0\}$ after each question word. This input text-location discrepancy between pre-training and DocVQA fine-tuning may also affect the VQA task performance.

**Explainable DocVQA with ROI Localization.** Albeit ViTLP underperforms the strong baseline DONUT, it acquires localization capacity to ground the answers, which is unprecedented to prior work. Benefiting from the layout localization capacity

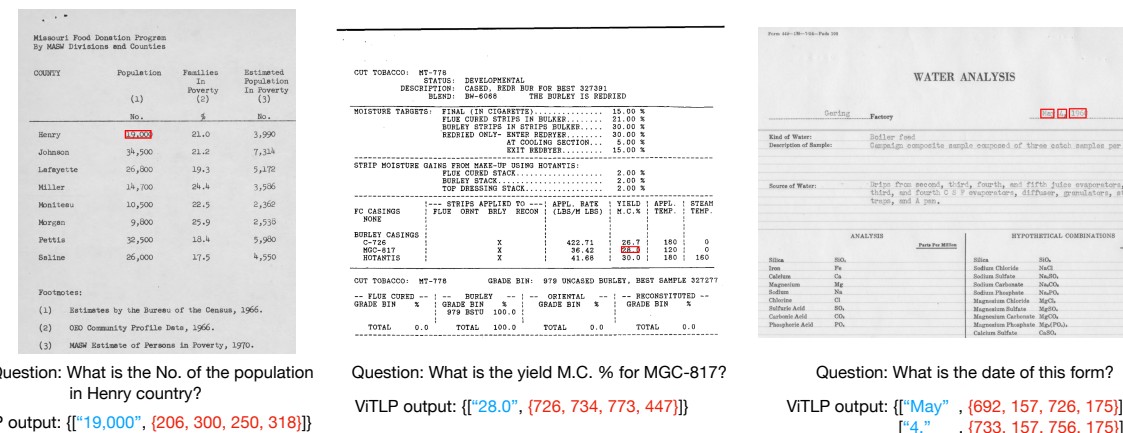

Figure 3: Visualization on ViTLP output answers for DocVQA. The ViTLP output sequences consist of answer words (in blue) and corresponding location coordinates (in red). For comprehensive visualization, we draw the region of interest (ROI) **referring to the output location coordinates** on the image.

learned in the pre-training phase, ViTLP can output the region of interest (ROI) with the generated answers, as presented in Figure 3. The visualized ROI localization can help users justify the model output answers, making the generative question-answering process more explainable and reliable. A potential application is to use ViTLP as a semi-automatic annotator to construct large-scale document answering datasets, where human annotators can easily justify the annotations according to the visualized ROI outputs.

## 4 Related Work

Visual document processing with multimodal pre-training has been widely studied recently. Depending on the pre-processing of documents, existing VDU works can be generally divided into two strands of research as listed below.

**OCR-based Methods.** Most existing VDU efforts (Xu et al., 2020, 2021; Huang et al., 2022; Peng et al., 2022; Li et al., 2021b; Bai et al., 2023) adopt OCR tools to localize and recognize document layouts and texts, and then feed them to the multimodal pre-trained models (Xu et al., 2021; Peng et al., 2022; Appalaraju et al., 2021; Li et al., 2021a). These methods usually involve multiple multimodal pre-training objectives over the vision, text, and layout. For instance, document words (Xu et al., 2020, 2021), textlines (Bai et al., 2023), regions (Li et al., 2021b; Wang et al., 2022b) are rich in document structure information to align visual features with text embeddings. Though promising, such pipeline suffers from heavy OCR pre-processing. Moreover, OCR errors can easily lead

to incorrect results for downstream tasks such as document question answering (Kim et al., 2022).

**OCR-free Methods.** There are few recent studies (Kim et al., 2022; Davis et al., 2022; Lee et al., 2023b) that jointly consider text recognition and understanding without OCR tools. For instance, Kim et al. (2022) takes document images as input to the network without prerequisite OCRs and conducts visual language pre-training. Lee et al. (2023b) further improves the pre-training objectives over various collected visual document corpus.

Our research lies within the OCR-free branch. Different from existing works, we first study to learning both document text and layout information from the input images. Our empirical results also verify that layout information not only enhances the learned representations for downstream VDU tasks but also makes the generation interpretable.

## 5 Conclusion

The paper proposes the visually guided generative text-layout pre-training (ViTLP) approach to boost the performance of visual document processing. The ViTLP model optimizes hierarchical language and layout modeling objectives to generate a mixed target sequence of texts and layouts. ViTLP can also serve as an OCR tool to locate and recognize texts of document images. Besides, the proposed approach can handle documents with arbitrary lengths via multi-segment generative pre-training. Experiments show that ViTLP achieves superior performance compared to existing counterparts on various VDU tasks.

## Limitations

Our community has entered the era of large language models (LLMs) (OpenAI, 2023; Chen et al., 2023). However, regarding the model size, ViTLP is still a rather small-scale pre-trained model, which limits its potential to become an interactive and generalized document AI assistant. In future work, we plan to explore two paths: 1) scaling up ViTLP with more parameters and training data, extending it to a more powerful foundation document model; 2) integrating ViTLP's text-layout aligned image encoder with open-source LLMs and instruction tuning (Zhu et al., 2023; Liu et al., 2023) to construct an interactive document AI system.

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

| Dataset | Size | Proportion | Document Type |
|---------|------|-----------|---------------|
| IIT-CDIP | $10,816,672$ | $81.89\%$ | Scanned Document |
| SynthDog | $2,000,000$ | $15.14\%$ | Synthetic Document |
| PublayNet | $261,076$ | $1.98\%$ | Scientific Paper |
| DocBank | $125,815$ | $0.95\%$ | Arxiv Paper |
| IAM | $1,198$ | $0.01\%$ | Hand Written |
| SciTSR | $3,536$ | $0.03\%$ | Figure and Table |

Table 5: Pre-training dataset statistics.

## A    Experiment Details

### A.1    Pre-training Data Statistics

Table 5 shows the pre-training data statistics. Following previous work, e.g., LayoutLMv2 (Xu et al., 2021) and FormNetv2 (Lee et al., 2023a), we use 11M IIT-CDIP document images as the main pre-training data. Besides, we follow Kim et al. (2022) and Davis et al. (2022) to include 2M machine-rendered synthetic documents for generative pre-training. Specifically, we adapt the official Syn­thDog generator[6] to generate synthetic document images with text and layout metadata. The other four corpora, i.e., IAM, SciTSR, PublayNet, and DocBank, account for only $\sim 3\%$ pre-training data whereby we aim to improve the diversity of pre-training document types.

The distribution of document sequence length is displayed in Figure 4. The number of text-layout sequence tokens follows a *long-tailed distribution*: there exist few long documents with the sequence length ranging from 1024 to 3072. This brings a trade-off to pre-training. With a relatively short sequence length (e.g., 512 tokens in LayoutLM), language modeling on long documents is incomplete, as the sequence tokens are truncated and wasted. However, with a relatively long sequence length (e.g., 3072), the GPU computation and memory overload would become prohibitive, which further forbids large batch sizes for better performance.[7] The multi-segment pre-training scheme can circumvent this bitter trade-off. Notably, the multi-segment processing scheme can be directly applied to long document fine-tuning (and inference). For example in the OCR and sequence labeling tasks, ViTLP also employs the multi-segment scheme to process the long documents by multiple segments with prefix context tokens.

---

[6]https://github.com/clovaai/donut/tree/master/synthdog

[7]Even assuming sufficient GPU resources, the long-tailed sequence-length distribution would also cause enormous token padding on long Transformer sequence inputs, leading to considerable waste of computational resources.

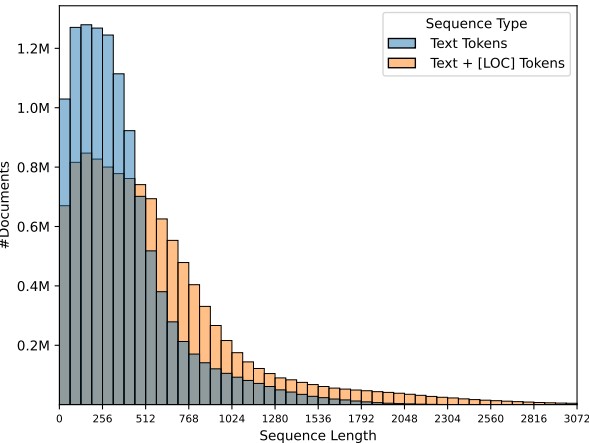

Figure 4: Distribution of document sequence length.

## B    Implementation Details of Sequential Layout Head

Given that multimodal interaction is learned by the stacked Transformer text-layout decoder layers, the LM and layout heads hereby function as a prober to output the next word and coordinate predictions. As introduced in Sec 2.2.2, the layout head predicts output probability $\text{Prob}(\mathbf{L}_{i,j})$ of the four coordinates $\{\mathbf{L}_{i,j}\}_{j=1}^{4} = \{z_{x1}, z_{y1}, z_{x2}, z_{y2}\}_i$ based on the $i$-th global [LOC] token's final hidden state $\mathbf{H}_{i,0} = \mathbf{H}_i^{VTL} \in \mathbb{R}^d$ as follows.

$$
\begin{cases}
\mathbf{H}_{i,1} = \text{GELU}\big(\mathbf{W}_s\mathbf{H}_{i,0}\big) \\
\mathbf{H}_{i,2} = \text{GELU}\big(\mathbf{W}_s\mathbf{H}_{i,1} + \text{E}_x'(\mathbf{L}_{i,1})\big) \\
\mathbf{H}_{i,3} = \text{GELU}\big(\mathbf{W}_s\mathbf{H}_{i,2} + \text{E}_y'(\mathbf{L}_{i,2})\big) \\
\mathbf{H}_{i,4} = \text{GELU}\big(\mathbf{W}_s\mathbf{H}_{i,3} + \text{E}_x'(\mathbf{L}_{i,3})\big)
\end{cases}
$$

$$\text{Prob}(\mathbf{L}_{i,j}) = \text{Softmax}\big(\mathbf{W}_L\mathbf{H}_{i,j}\big)$$

The coordinate tokens are quantized into a discrete range of $[0, 1000]$, making the layout-token vocabulary size of $|L| = 1001$. The layout head parameters are lightweight including a hidden matrix $\mathbf{W}_s \in \mathbb{R}^{d \times d}$, two embeddings $\text{E}_x'(\cdot) \in \mathbb{R}^d$ and $\text{E}_y'(\cdot) \in \mathbb{R}^d$, and a linear projection $\mathbf{W}_L \in \mathbb{R}^{|L| \times d}$. We use the same GELU activation (Hendrycks and Gimpel, 2016) as in the Transformer layers. The layout head works similar to a vanilla RNN, as each coordinate decoding step also considers the information of previous coordinates. Compared to naively using four independent linear heads, the sequential layout head can capture the spatial relation among the output coordinates (e.g., $x_1 < x_2$ and $y_1 < y_2$), bootstrapping more accurate coordinate prediction in inference.

## C Qualitative Cases of ViTLP Document OCR Functionality

As shown in Figure 5 to 7, ViTLP demonstrates its functionality on zero-shot document OCR. ViTLP outputs the word-level OCR results including texts and bounding boxes.

# Attention Is All You Need

**Ashish Vaswani***
Google Brain
avaswani@google.com

**Noam Shazeer***
Google Brain
noam@google.com

**Niki Parmar***
Google Research
nikip@google.com

**Jakob Uszkoreit***
Google Research
usz@google.com

**Llion Jones***
Google Research
llion@google.com

**Aidan N. Gomez*** [†]
University of Toronto
aidan@cs.toronto.edu

**Łukasz Kaiser***
Google Brain
lukaszkaiser@google.com

**Illia Polosukhin*** [‡]
illia.polosukhin@gmail.com

## Abstract

The dominant sequence transduction models are based on complex recurrent or convolutional neural networks that include an encoder and a decoder. The best performing models also connect the encoder and decoder through an attention mechanism. We propose a new simple network architecture, the Transformer, based solely on attention mechanisms, dispensing with recurrence and convolutions entirely. Experiments on two machine translation tasks show these models to be superior in quality while being more parallelizable and requiring significantly less time to train. Our model achieves 28.4 BLEU on the WMT 2014 English-to-German translation task, improving over the existing best results, including ensembles, by over 2 BLEU. On the WMT 2014 English-to-French translation task, our model establishes a new single-model state-of-the-art BLEU score of 41.0 after training for 3.5 days on eight GPUs, a small fraction of the training costs of the best models from the literature.

## 1  Introduction

Recurrent neural networks, long short-term memory [12] and gated recurrent [7] neural networks in particular, have been firmly established as state of the art approaches in sequence modeling and transduction problems such as language modeling and machine translation [29, 2, 5]. Numerous efforts have since continued to push the boundaries of recurrent language models and encoder-decoder architectures [31, 21, 13].

---

*Equal contribution. Listing order is random. Jakob proposed replacing RNNs with self-attention and started the effort to evaluate this idea. Ashish, with Illia, designed and implemented the first Transformer models and has been crucially involved in every aspect of this work. Noam proposed scaled dot-product attention, multi-head attention and the parameter-free position representation and became the other person involved in nearly every detail. Niki designed, implemented, tuned and evaluated countless model variants in our original codebase and tensor2tensor. Llion also experimented with novel model variants, was responsible for our initial codebase, and efficient inference and visualizations. Lukasz and Aidan spent countless long days designing various parts of and implementing tensor2tensor, replacing our earlier codebase, greatly improving results and massively accelerating our research.

†Work performed while at Google Brain.
‡Work performed while at Google Research.

31st Conference on Neural Information Processing Systems (NIPS 2017), Long Beach, CA, USA.

Figure 5: ViTLP OCR results on a paper. For comprehensive visualization, we draw the output texts (in blue) and bounding boxes (in red) according to the OCR outputs.

```
1    "Attention", [306, 47, 456, 77]
2    "Is", [465, 47, 493, 77]
3    "All", [502, 47, 550, 77]
4    "You", [559, 47, 619, 77]
5    "Need", [628, 47, 708, 77]
6    "Ashish", [139, 174, 201, 191]
7    "Vaswani*", [206, 174, 290, 191]
8    "Noam", [365, 174, 420, 191]
9    "Shazeer*", [425, 174, 505, 191]
10   "Niki", [578, 174, 617, 191]
11   "Parmar*", [622, 174, 701, 191]
12   "Jakob", [761, 174, 815, 191]
13   "Uszkoreit*", [821, 174, 915, 191]
14   "Google", [152, 191, 215, 207]
15   "Brain", [221, 191, 268, 207]
16   "Google", [373, 191, 436, 207]
17   "Brain", [441, 191, 489, 207]
18   "Google", [562, 191, 625, 207]
19   "Research", [630, 191, 708, 207]
20   "Google", [760, 191, 823, 207]
21   "Research", [829, 191, 907, 207]
22   "avaswani@google.com", [104, 207, 317, 223]
23   "noam@google.com", [346, 207, 515, 223]
24   "nikipl@google.com", [545, 207, 725, 223]
25   "usz@google.com", [755, 207, 912, 223]
26   "Llion", [162, 246, 212, 264]
27   "Jones*", [217, 246, 275, 264]
28   "Aidan", [386, 246, 442, 264]
29   "N.", [447, 246, 468, 264]
30   "Gomez*", [473, 246, 546, 264]
31   "\u2020", [552, 244, 560, 260]
32   "Lukasz", [696, 246, 762, 264]
33   "Kaiser*", [767, 246, 836, 264]
34   "Google", [142, 263, 205, 280]
35   "Research", [210, 263, 288, 280]
36   "University", [377, 263, 466, 280]
37   "of", [471, 263, 489, 280]
38   "Toronto", [494, 263, 561, 280]
39   "Google", [704, 263, 767, 280]
40   "Brain", [772, 263, 820, 280]
41   "llion@google.com", [126, 279, 305, 295]
42   "aidan@cs.toronto.edu", [357, 279, 580, 295]
43   "lukaszkaiser@google.com", [634, 279, 890, 295]
44   "Illia", [429, 318, 465, 336]
45   "Polosukhin*", [471, 318, 581, 336]
46   "\u2021", [587, 316, 595, 333]
47   "illia.polosukhin@gmail.com", [364, 336, 653, 352]
48   "Abstract", [460, 391, 555, 413]
49   "The", [162, 426, 196, 443]
50   "dominant", [202, 426, 284, 443]
51   "sequence", [290, 426, 369, 443]
52   "transduction", [375, 426, 482, 443]
53   "models", [488, 426, 551, 443]
54   "are", [556, 426, 583, 443]
55   "based", [589, 426, 639, 443]
56   "on", [644, 426, 666, 443]
57   "complex", [672, 426, 746, 443]
58   "recurrent", [752, 426, 830, 443]
59   "or", [836, 426, 855, 443]
60   "convolutional", [162, 443, 280, 460]
61   "neural", [286, 443, 341, 460]
62   "networks", [347, 443, 426, 460]
63   "that", [432, 443, 465, 460]
64   "include", [471, 443, 535, 460]
65   "an", [541, 443, 562, 460]
66   "encoder", [568, 443, 637, 460]
67   "and", [643, 443, 675, 460]
68   "a", [681, 443, 691, 460]
69   "decoder.", [697, 443, 770, 460]
70   "The", [779, 443, 813, 460]
71   "best", [819, 443, 854, 460]
72   "performing", [162, 459, 260, 475]
73   "models", [266, 459, 330, 475]
74   "also", [336, 459, 372, 475]
75   "connect", [378, 459, 446, 475]
76   "the", [453, 459, 480, 475]
77   "encoder", [486, 459, 555, 475]
78   "and", [562, 459, 594, 475]
79   "decoder", [600, 459, 669, 475]
80   "through", [676, 459, 744, 475]
81   "an", [750, 459, 771, 475]
82   "attention", [778, 459, 854, 475]
83   "mechanism.", [162, 475, 267, 491]
84   "We", [279, 475, 308, 491]
85   "propose", [315, 475, 384, 491]
86   "a", [391, 475, 401, 491]
87   "new", [408, 475, 444, 491]
88   "simple", [451, 475, 509, 491]
89   "network", [516, 475, 587, 491]
90   "architecture,", [594, 475, 703, 491]
91   "the", [710, 475, 737, 491]
92   "Transformer,", [744, 475, 856, 491]
93   "based", [162, 491, 210, 507]
94   "solely", [215, 491, 264, 507]
95   "on", [269, 491, 290, 507]
96   "attention", [295, 491, 367, 507]
97   "mechanisms,", [372, 491, 480, 507]
98   "dispensing", [484, 491, 574, 507]
99   "with", [579, 491, 615, 507]
100  "recurrence", [620, 491, 708, 507]
101  "and", [713, 491, 743, 507]
102  "convolutions", [747, 491, 854, 507]
103  "entirely.", [162, 506, 233, 523]
104  "Experiments", [245, 506, 355, 523]
105  "on", [363, 506, 385, 523]
106  "two", [392, 506, 424, 523]
107  "machine", [431, 506, 505, 523]
108  "translation", [512, 506, 604, 523]
109  "tasks", [611, 506, 655, 523]
110  "show", [662, 506, 708, 523]
111  "these", [715, 506, 760, 523]
112  "models", [767, 506, 830, 523]
113  "to", [837, 506, 854, 523]
114  "be", [162, 522, 182, 538]
115  "superior", [188, 522, 258, 538]
116  "in", [264, 522, 280, 538]
117  "quality", [286, 522, 346, 538]
118  "while", [351, 522, 398, 538]
119  "being", [404, 522, 451, 538]
120  "more", [457, 522, 501, 538]
121  "parallelizable", [506, 522, 622, 538]
122  "and", [628, 522, 659, 538]
123  "requiring", [664, 522, 743, 538]
124  "significantly", [748, 522, 854, 538]
125  "less", [162, 538, 195, 554]
126  "time", [201, 538, 240, 554]
127  "to", [246, 538, 263, 554]
128  "train.", [269, 538, 315, 554]
129  "Our", [325, 538, 359, 554]
130  "model", [365, 538, 419, 554]
131  "achieves", [426, 538, 500, 554]
132  "28.4", [506, 538, 544, 554]
133  "BLEU", [550, 538, 607, 554]
134  "on", [613, 538, 635, 554]
135  "the", [641, 538, 668, 554]
136  "WMT", [675, 538, 727, 554]
137  "2014", [733, 538, 778, 554]
138  "English-", [784, 538, 856, 554]
139  "to-German", [162, 554, 256, 570]
140  "translation", [263, 554, 355, 570]
141  "task,", [362, 554, 402, 570]
142  "improving", [409, 554, 500, 570]
143  "over", [507, 554, 545, 570]
144  "the", [552, 554, 578, 570]
145  "existing", [585, 554, 653, 570]
146  "best", [660, 554, 696, 570]
147  "results,", [703, 554, 764, 570]
148  "including", [772, 554, 854, 570]
149  "ensembles,", [162, 570, 254, 586]
150  "by", [259, 570, 280, 586]
151  "over", [285, 570, 322, 586]
152  "2", [329, 570, 336, 586]
153  "BLEU.", [341, 570, 401, 586]
154  "On", [406, 570, 431, 586]
155  "the", [436, 570, 462, 586]
156  "WMT", [467, 570, 517, 586]
157  "2014", [522, 570, 563, 586]
158  "English-to-French", [568, 570, 720, 586]
159  "translation", [725, 570, 813, 586]
160  "task,", [818, 570, 856, 586]
161  "our", [162, 586, 190, 602]
162  "model", [195, 586, 247, 602]
163  "establishes", [252, 586, 342, 602]
164  "a", [347, 586, 357, 602]
165  "new", [362, 586, 396, 602]
166  "single-model", [401, 586, 510, 602]
167  "state-of-the-art", [515, 586, 639, 602]
168  "BLEU", [644, 586, 698, 602]
169  "score", [703, 586, 747, 602]
170  "of", [752, 586, 770, 602]
171  "41.0", [775, 586, 811, 602]
172  "after", [816, 586, 855, 602]
173  "training", [162, 601, 230, 617]
174  "for", [236, 601, 261, 617]
175  "3.5", [267, 601, 293, 617]
176  "days", [299, 601, 339, 617]
177  "on", [345, 601, 366, 617]
178  "eight", [372, 601, 415, 617]
179  "GPUs,", [421, 601, 478, 617]
180  "a", [484, 601, 494, 617]
181  "small", [500, 601, 546, 617]
182  "fraction", [552, 601, 620, 617]
183  "of", [626, 601, 643, 617]
184  "the", [649, 601, 676, 617]
185  "training", [682, 601, 749, 617]
186  "costs", [755, 601, 798, 617]
187  "of", [804, 601, 822, 617]
188  "the", [641, 601, 854, 617]
189  "best", [162, 617, 197, 633]
190  "models", [202, 617, 263, 633]
191  "from", [269, 617, 310, 633]
192  "the", [315, 617, 341, 633]
193  "literature.", [346, 617, 428, 633]
194  "1", [85, 657, 98, 679]
195  "Introduction", [124, 657, 263, 679]
196  "Recurrent", [85, 694, 171, 711]
197  "neural", [177, 694, 231, 711]
198  "networks,", [237, 694, 321, 711]
199  "long", [327, 694, 366, 711]
200  "short-term", [372, 694, 462, 711]
201  "memory", [468, 694, 540, 711]
202  "[12]", [546, 694, 581, 711]
203  "and", [587, 694, 618, 711]
204  "gated", [624, 694, 671, 711]
205  "recurrent", [677, 694, 755, 711]
206  "[7]", [761, 694, 785, 711]
207  "neural", [791, 694, 845, 711]
208  "networks", [851, 694, 930, 711]
209  "in", [85, 710, 102, 727]
210  "particular,", [108, 710, 195, 727]
211  "have", [200, 710, 240, 727]
212  "been", [246, 710, 287, 727]
213  "firmly", [292, 710, 345, 727]
214  "established", [350, 710, 446, 727]
215  "as", [452, 710, 469, 727]
216  "state", [475, 710, 514, 727]
217  "of", [520, 710, 538, 727]
218  "the", [543, 710, 570, 727]
219  "art", [575, 710, 598, 727]
220  "approaches", [603, 710, 700, 727]
221  "in", [706, 710, 722, 727]
222  "sequence", [728, 710, 807, 727]
223  "modeling", [812, 710, 894, 727]
224  "and", [899, 710, 930, 727]
225  "transduction", [85, 726, 193, 743]
226  "problems", [199, 726, 280, 743]
227  "such", [286, 726, 326, 743]
228  "as", [332, 726, 351, 743]
229  "language", [357, 726, 435, 743]
230  "modeling", [441, 726, 523, 743]
231  "and", [529, 726, 561, 743]
232  "translation", [567, 726, 640, 743]
233  "translation", [647, 726, 738, 743]
234  "[29,", [745, 726, 778, 743]
235  "2,", [785, 726, 801, 743]
236  "5].", [808, 726, 830, 743]
237  "Numerous", [840, 726, 930, 743]
238  "efforts", [85, 742, 139, 759]
239  "have", [144, 742, 183, 759]
240  "since", [187, 742, 230, 759]
241  "continued", [234, 742, 317, 759]
242  "to", [321, 742, 337, 759]
243  "push", [342, 742, 381, 759]
244  "the", [386, 742, 411, 759]
245  "boundaries", [416, 742, 507, 759]
246  "of", [512, 742, 529, 759]
247  "recurrent", [534, 742, 609, 759]
248  "language", [614, 742, 689, 759]
249  "models", [693, 742, 753, 759]
250  "and", [758, 742, 788, 759]
251  "encoder-decoder", [793, 742, 930, 759]
252  "architectures", [85, 758, 194, 775]
253  "[31,", [199, 758, 233, 775]
254  "21,", [238, 758, 265, 775]
255  "13].", [271, 758, 303, 775]
256  "*Equal", [111, 781, 164, 797]
257  "contribution.", [169, 781, 265, 797]
258  "Listing", [270, 781, 324, 797]
259  "order", [329, 781, 368, 797]
260  "is", [373, 781, 385, 797]
261  "random.", [390, 781, 452, 797]
262  "Jakob", [459, 781, 502, 797]
263  "proposed", [507, 781, 576, 797]
264  "replacing", [580, 781, 650, 797]
265  "RNNs", [655, 781, 702, 797]
266  "with", [707, 781, 740, 797]
267  "self-attention", [745, 781, 844, 797]
268  "and", [849, 781, 876, 797]
269  "started", [880, 781, 931, 797]
270  "the", [85, 796, 109, 812]
271  "effort", [114, 796, 156, 812]
272  "to", [161, 796, 176, 812]
273  "evaluate", [181, 796, 244, 812]
274  "this", [249, 796, 277, 812]
275  "idea.", [281, 796, 318, 812]
276  "Ashish,", [324, 796, 383, 812]
277  "with", [388, 796, 422, 812]
278  "Illia,", [426, 796, 462, 812]
279  "designed", [466, 796, 536, 812]
280  "and", [540, 796, 568, 812]
281  "implemented,", [573, 796, 674, 812]
282  "the", [678, 796, 702, 812]
283  "first", [707, 796, 737, 812]
284  "Transformer", [742, 796, 838, 812]
285  "models", [842, 796, 898, 812]
286  "and", [902, 796, 930, 812]
287  "has", [85, 810, 111, 826]
288  "been", [115, 810, 151, 826]
289  "crucially", [155, 810, 220, 826]
290  "involved", [224, 810, 289, 826]
291  "in", [293, 810, 308, 826]
292  "every", [312, 810, 353, 826]
293  "aspect", [357, 810, 404, 826]
294  "of", [408, 810, 424, 826]
295  "this", [428, 810, 455, 826]
296  "work.", [459, 810, 503, 826]
297  "Noam", [508, 810, 554, 826]
298  "proposed", [559, 810, 628, 826]
299  "scaled", [632, 810, 679, 826]
300  "dot-product", [683, 810, 770, 826]
300  "dot-product", [683, 810, 770, 826]
301  "attention,", [775, 810, 845, 826]
302  "multi-head", [849, 810, 930, 826]
303  "attention", [85, 825, 153, 841]
304  "and", [158, 825, 186, 841]
305  "the", [191, 825, 214, 841]
306  "parameter-free", [219, 825, 333, 841]
307  "position", [338, 825, 400, 841]
308  "representation", [405, 825, 514, 841]
309  "and", [519, 825, 547, 841]
310  "became", [552, 825, 610, 841]
311  "the", [615, 825, 639, 841]
312  "other", [644, 825, 684, 841]
313  "person", [689, 825, 740, 841]
314  "involved", [745, 825, 811, 841]
315  "in", [816, 825, 831, 841]
316  "nearly", [836, 825, 884, 841]
317  "every", [889, 825, 930, 841]
318  "detail.", [85, 839, 133, 855]
319  "Niki", [138, 839, 173, 855]
320  "designed,", [178, 839, 250, 855]
321  "implemented,", [255, 839, 359, 855]
322  "tuned", [364, 839, 407, 855]
323  "and", [412, 839, 439, 855]
324  "evaluated", [444, 839, 516, 855]
325  "countless", [521, 839, 592, 855]
326  "model", [597, 839, 644, 855]
327  "variants", [649, 839, 709, 855]
328  "in", [714, 839, 729, 855]
329  "our", [734, 839, 759, 855]
330  "original", [764, 839, 823, 855]
331  "codebase", [828, 839, 898, 855]
332  "and", [903, 839, 930, 855]
333  "tensor2tensor.", [85, 854, 192, 869]
334  "Llion", [197, 854, 239, 869]
335  "also", [243, 854, 274, 869]
336  "experimented", [278, 854, 382, 869]
337  "with", [386, 854, 420, 869]
338  "novel", [425, 854, 467, 869]
339  "model", [471, 854, 519, 869]
340  "variants,", [524, 854, 588, 869]
341  "was", [593, 854, 622, 869]
342  "responsible", [627, 854, 713, 869]
343  "for", [718, 854, 740, 869]
344  "our", [745, 854, 770, 869]
345  "initial", [775, 854, 819, 869]
346  "codebase,", [824, 854, 898, 869]
347  "and", [903, 854, 930, 869]
348  "efficient", [85, 869, 147, 884]
349  "inference", [151, 869, 221, 884]
350  "and", [225, 869, 253, 884]
351  "visualizations.", [257, 869, 365, 884]
352  "Lukasz", [370, 869, 424, 884]
353  "and", [429, 869, 456, 884]
354  "Aidan", [461, 869, 506, 884]
355  "spent", [511, 869, 551, 884]
356  "countless", [555, 869, 625, 884]
357  "long", [629, 869, 665, 884]
358  "days", [667, 869, 701, 884]
359  "designing", [706, 869, 779, 884]
360  "various", [783, 869, 838, 884]
361  "parts", [842, 869, 879, 884]
362  "of", [883, 869, 899, 884]
363  "and", [903, 869, 930, 884]
364  "implementing", [85, 883, 190, 898]
365  "tensor2tensor,", [194, 883, 299, 898]
366  "replacing", [303, 883, 374, 898]
367  "our", [378, 883, 403, 898]
368  "earlier", [407, 883, 455, 898]
369  "codebase,", [460, 883, 533, 898]
370  "greatly", [537, 883, 590, 898]
371  "improving", [594, 883, 672, 898]
372  "results,", [676, 883, 725, 898]
373  "and", [730, 883, 757, 898]
374  "massively", [761, 883, 836, 898]
375  "accelerating", [840, 883, 930, 898]
376  "our", [85, 898, 111, 913]
377  "research.", [116, 898, 184, 913]
378  "\u2020Work", [112, 911, 162, 928]
379  "performed", [167, 913, 247, 928]
380  "while", [252, 913, 294, 928]
381  "at", [299, 913, 313, 928]
382  "Google", [317, 913, 374, 928]
383  "Brain.", [378, 913, 426, 928]
384  "\u2021Work", [112, 927, 162, 944]
385  "performed", [167, 929, 247, 944]
386  "while", [252, 929, 294, 944]
387  "at", [299, 929, 313, 944]
388  "Google", [317, 929, 374, 944]
389  "Research.", [378, 929, 453, 944]
390  "3", [86, 967, 94, 983]
391  "1st", [98, 967, 118, 983]
392  "Conference", [123, 967, 210, 983]
393  "on", [215, 967, 235, 983]
394  "Neural", [239, 967, 292, 983]
395  "Information", [296, 967, 388, 983]
396  "Processing", [392, 967, 475, 983]
397  "Systems", [480, 967, 543, 983]
398  "(NIPS", [548, 967, 596, 983]
399  "2017),", [601, 967, 650, 983]
400  "Long", [655, 967, 695, 983]
401  "Beach,", [700, 967, 753, 983]
402  "CA,", [758, 967, 789, 983]
403  "USA.", [794, 967, 836, 983]
```

Figure 6: ViTLP OCR results as visualized in Figure 5 above.

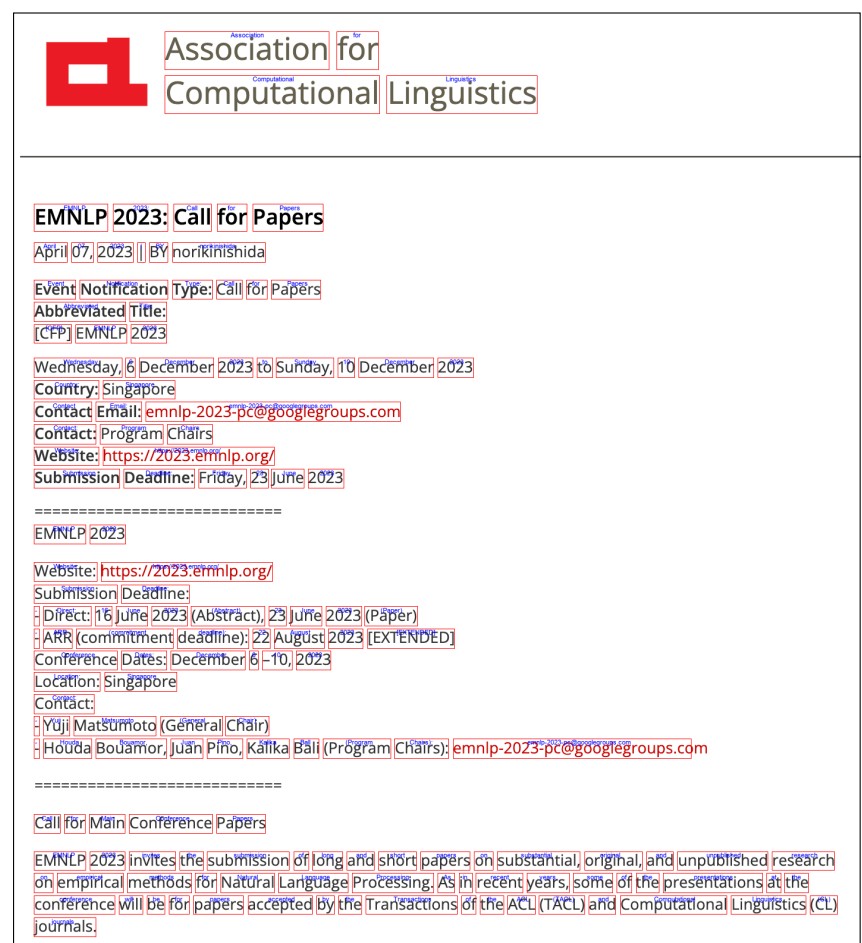

```
1   "Association", [179, 20, 373, 61]          41  "Contact:", [25, 441, 98, 462]           81   "2023", [334, 684, 375, 704]                      121  "on", [25, 922, 47, 943]
2   "for", [382, 20, 431, 61]                  42  "Program", [103, 441, 177, 462]          82   "Location:", [25, 707, 103, 727]                  122  "empirical", [52, 922, 130, 943]
3   "Computational", [179, 67, 432, 108]       43  "Chairs", [182, 441, 235, 462]           83   "Singapore", [108, 707, 193, 727]                 123  "methods", [135, 922, 211, 943]
4   "Linguistics", [441, 67, 618, 108]         44  "Website:", [25, 465, 101, 485]          84   "Contact:", [25, 730, 96, 751]                    124  "for", [215, 922, 240, 943]
5   "EMNLP", [25, 205, 112, 234]               45  "https://2023.emnlp.org/", [106, 465, 308, 485]   85   "-", [25, 754, 31, 775]                  125  "Natural", [245, 922, 308, 943]
6   "2023:", [118, 205, 183, 234]              46  "Submission", [25, 489, 126, 509]        86   "Yuji", [36, 754, 67, 775]                        126  "Language", [313, 922, 395, 943]
7   "Call", [190, 205, 234, 234]               47  "Deadline:", [131, 489, 214, 509]        87   "Matsumoto", [72, 754, 169, 775]                  127  "Processing.", [400, 922, 496, 943]
8   "for", [241, 205, 276, 234]                48  "Friday,", [219, 489, 275, 509]          88   "(General", [174, 754, 247, 775]                  128  "As", [501, 922, 521, 943]
9   "Papers", [283, 205, 366, 234]             49  "23", [280, 489, 301, 509]               89   "Chair)", [251, 754, 302, 775]                    129  "in", [526, 922, 542, 943]
10  "April", [25, 246, 64, 267]                50  "June", [305, 489, 343, 509]             90   "-", [25, 778, 31, 799]                           130  "recent", [546, 922, 601, 943]
11  "07,", [69, 246, 95, 267]                  51  "2023", [348, 489, 390, 509]             91   "Houda", [36, 778, 93, 799]                       131  "years,", [606, 922, 656, 943]
12  "2023", [100, 246, 142, 267]               52  "EMNLP", [25, 549, 86, 569]              92   "Bouamor,", [98, 778, 181, 799]                   132  "some", [661, 922, 707, 943]
13  "|", [147, 246, 156, 267]                  53  "2023", [91, 549, 133, 569]              93   "Juan", [186, 778, 224, 799]                      133  "of", [712, 922, 730, 943]
14  "BY", [161, 246, 183, 267]                 54  "Website:", [25, 589, 99, 609]           94   "Pino,", [229, 778, 271, 799]                     134  "the", [735, 922, 762, 943]
15  "norikinishida", [188, 246, 298, 267]      55  "https://2023.emnlp.org/", [104, 589, 306, 609]   95   "Kalika", [276, 778, 326, 799]          135  "presentations", [767, 922, 884, 943]
16  "Event", [25, 286, 74, 307]                56  "Submission", [25, 613, 123, 633]        96   "Ball", [331, 778, 361, 799]                      136  "at", [889, 922, 906, 943]
17  "Notification", [79, 286, 183, 307]        57  "Deadline:", [128, 613, 208, 633]        97   "(Program", [366, 778, 446, 799]                  137  "the", [910, 922, 938, 943]
18  "Type:", [188, 286, 235, 307]              58  "-", [25, 636, 31, 657]                  98   "Chairs):", [450, 778, 514, 799]                  138  "conference", [25, 946, 120, 966]
19  "Call", [240, 286, 270, 307]               59  "Direct:", [36, 636, 92, 657]            99   "emnlp-2023-pc@googlegroups.com", [519, 778, 800, 799]   139  "will", [125, 946, 153, 966]
20  "for", [275, 286, 300, 307]                60  "16", [97, 636, 117, 657]                100  "Call", [25, 859, 56, 879]                        140  "be", [158, 946, 179, 966]
21  "Papers", [305, 286, 363, 307]             61  "June", [122, 636, 159, 657]             101  "for", [61, 859, 85, 879]                         141  "for", [184, 946, 208, 966]
22  "Abbreviated", [25, 310, 133, 331]         62  "2023", [164, 636, 206, 657]             102  "Main", [90, 859, 132, 879]                       142  "papers", [213, 946, 272, 966]
23  "Title:", [138, 310, 181, 331]             63  "(Abstract),", [210, 636, 297, 657]      103  "Conference", [137, 859, 235, 879]                143  "accepted", [277, 946, 353, 966]
24  "[CFP]", [25, 334, 69, 354]                64  "23", [302, 636, 322, 657]               104  "Papers", [240, 859, 298, 879]                    144  "by", [358, 946, 379, 966]
25  "EMNLP", [74, 334, 134, 354]               65  "June", [327, 636, 364, 657]             105  "EMNLP", [25, 899, 86, 919]                       145  "the", [384, 946, 411, 966]
26  "2023", [139, 334, 181, 354]               66  "2023", [369, 636, 411, 657]             106  "2023", [91, 899, 132, 919]                       146  "Transactions", [416, 946, 524, 966]
27  "Wednesday,", [25, 370, 129, 391]          67  "(Paper)", [415, 636, 477, 657]          107  "invites", [137, 899, 192, 919]                   147  "of", [529, 946, 546, 966]
28  "6", [133, 370, 144, 391]                  68  "-", [25, 660, 31, 681]                  108  "the", [197, 899, 225, 919]                       148  "the", [551, 946, 579, 966]
29  "December,", [149, 370, 237, 391]          69  "ARR", [36, 660, 70, 681]                109  "submission", [229, 899, 326, 919]                149  "ACL", [583, 946, 616, 966]
30  "2023", [242, 370, 283, 391]               70  "(commitment", [75, 660, 189, 681]       110  "of", [331, 899, 349, 919]                        150  "(TACL)", [620, 946, 674, 966]
31  "to", [288, 370, 306, 391]                 71  "deadline):", [194, 660, 278, 681]       111  "long", [353, 899, 389, 919]                      151  "and", [679, 946, 711, 966]
32  "Sunday,", [310, 370, 378, 391]            72  "22", [283, 660, 303, 681]               112  "and", [394, 899, 426, 919]                       152  "Computational", [716, 946, 842, 966]
33  "10", [383, 370, 403, 391]                 73  "August", [308, 660, 367, 681]           113  "short", [431, 899, 476, 919]                     153  "Linguistics", [847, 946, 935, 966]
34  "December,", [408, 370, 496, 391]          74  "2023", [372, 660, 413, 681]             114  "papers", [481, 899, 539, 919]                    154  "(CL)", [940, 946, 972, 966]
35  "2023", [501, 370, 543, 391]               75  "[EXTENDED]", [418, 660, 521, 681]       115  "on", [544, 899, 566, 919]                        155  "journals.", [25, 970, 98, 990]
36  "Country:", [25, 394, 101, 414]            76  "Conference", [25, 684, 123, 704]        116  "substantial,", [571, 899, 669, 919]
37  "Singapore", [106, 394, 191, 414]          77  "Dates:", [128, 684, 181, 704]           117  "original,", [674, 899, 742, 919]
38  "Contact", [25, 417, 93, 438]              78  "December", [186, 684, 274, 704]         118  "and", [747, 899, 779, 919]
39  "Email:", [98, 417, 152, 438]              79  "6", [279, 684, 289, 704]                119  "unpublished", [784, 899, 890, 919]
40  "emnlp-2023-pc@googlegroups.com", [157, 417, 457, 438]   80  "-10,", [294, 684, 329, 704]   120  "research", [895, 899, 969, 919]
```

Figure 7: ViTLP OCR results on a webpage. For comprehensive visualization, we draw the output texts (in blue) and bounding boxes (in red) according to the beneath ViTLP OCR outputs.