# OpenReview forum: "Visually Guided Generative Text-Layout Pre-training for Document Intelligence"
_EMNLP/2023/Conference — Submitted to EMNLP 2023_

### Official Review · Reviewer_5J31 · 2023-08-05

**Soundness:** 2

**Excitement:**

3: Ambivalent: It has merits (e.g., it reports state-of-the-art results, the idea is nice), but there are key weaknesses (e.g., it describes incremental work), and it can significantly benefit from another round of revision. However, I won't object to accepting it if my co-reviewers champion it.

**Missing References:**

Tang, Zineng et al. “Unifying Vision, Text, and Layout for Universal Document Processing.” CVPR 2023

**Paper Topic And Main Contributions:**

This paper proposes a novel pre-training method called Visually Guided generative Text-Layout Pre-training (ViTLP) for visual document understanding. The key contributions are:
1. ViTLP jointly models text and layout information from document images in a generative fashion, without the need for OCR preprocessing. This is achieved by optimizing objectives for hierarchical language and layout modeling to generate mixed sequences of text and layout.
2. ViTLP can process long documents through a multi-segment pre-training scheme, breaking the input into segments and using previous tokens as context to generate subsequent tokens.
3. ViTLP can serve as an end-to-end OCR model for text localization and recognition. It also supports various downstream VDU tasks by taking task-specific prefixes as input.
4. Experiments show ViTLP achieves state-of-the-art results on benchmark VDU datasets, outperforming prior works. It also provides interpretable outputs with localized text regions.

**Reasons To Accept:**

1. Unified pre-training - ViTLP can serve as both an OCR engine and a backbone for downstream VDU, enabling unified document intelligence.
2. Explainability - The generated layout provides interpretable ROI visualization for text-driven tasks like QA.

**Reasons To Reject:**

1. The literature review is incomplete - the paper does not discuss and compare with some highly relevant recent works like LiLT, LayoutLMv3, and UDOP[1]. These methods are not compared in the experiments tables. Adding experiments on these models would better showcase the performance of ViTLP.

2. This paper can benefit from including benchmarks like InfographicVQA. The authors propose an OCR-free approach where visual information plays a crucial role. InfographicVQA is a benchmark similar to DocVQA but requires models to understand more complex and visually rich documents that include informative charts and posters. This benchmark has been used in previous document understanding research.

3. Although the model performs well in document classification and extraction tasks, its performance is relatively lower compared to previous methods on more challenging question-answering task DovVQA.

4. Reproducibility concerns - implementation details are brief, so reproducing the reported results could be difficult. More info like hyper-parameters and training details would increase reproducibility.

[1] Tang, Zineng et al. “Unifying Vision, Text, and Layout for Universal Document Processing.” CVPR 2023

**Reproducibility:**

3: Could reproduce the results with some difficulty. The settings of parameters are underspecified or subjectively determined; the training/evaluation data are not widely available.

**Reviewer Confidence:**

4: Quite sure. I tried to check the important points carefully. It's unlikely, though conceivable, that I missed something that should affect my ratings.

---

> ### Author Rebuttal · Authors · 2023-08-29
>
> **Q1: The literature review is incomplete - the paper does not discuss and compare with some highly relevant recent works like LiLT, LayoutLMv3, and UDOP[1]. These methods are not compared in the experiments tables. Adding experiments on these models would better showcase the performance of ViTLP.**
>
> **Reply:** Thanks for your valuable suggestion. We will add the comparison to LiLT, LayoutLMv3, and UDOP to the revised manuscript.
>
> **Q2: This paper can benefit from including benchmarks like InfographicVQA. The authors propose an OCR-free approach where visual information plays a crucial role. InfographicVQA is a benchmark similar to DocVQA but requires models to understand more complex and visually rich documents that include informative charts and posters. This benchmark has been used in previous document understanding research.**
>
> **Reply:** As we follow LayoutLM-v2 and DONUT to perform benchmark evaluation, we did not consider this benchmark before. We will add results on InfographicVQA in the revised manuscript.
>
>
> **Q3: Although the model performs well in document classification and extraction tasks, its performance is relatively lower compared to previous methods on more challenging question-answering task DovVQA.**
>
> **Reply:** Yes, ViTLP did not achieve SOTA performance in some VDU tasks. However, to the best of our knowledge, ViTLP is the first general document model to perform the tasks of OCR, document understanding (FUNSD, CORD, and RVL-CDIP), and generative document VQA (with explainable ROI outputs). Besides, ViTLP is able to process long documents and generate layout-level outputs. Though ViTLP could not achieve SOTA performance on some tasks (e.g., underperform DONUT [1] and UDOP [2]), we hope the reviewer can take the model’s general functionalities and performance on various tasks into account.
>
>
>
> **Q4: Reproducibility concerns - implementation details are brief, so reproducing the reported results could be difficult. More info like hyper-parameters and training details would increase reproducibility.**
>
> **Reply:** We list the important implementation details including pre-training hyper-parameters in Lines 313 - 343. It would be helpful if the reviewer let us know what implementation details are missed in the discussion period, so that we can address the reviewer’s concern and add the details to the revised manuscript.
>
>
>
> [1] OCR-free Document Understanding Transformer, ECCV 2022.
>
> [2] Unifying Vision, Text, and Layout for Universal Document Processing, CVPR 2023.

---

### Official Review · Reviewer_BkUh · 2023-08-06

**Soundness:** 2

**Excitement:**

3: Ambivalent: It has merits (e.g., it reports state-of-the-art results, the idea is nice), but there are key weaknesses (e.g., it describes incremental work), and it can significantly benefit from another round of revision. However, I won't object to accepting it if my co-reviewers champion it.

**Missing References:**

The tasks of visual document analysis are not fully reviewed. It should be noted that pre-trained models should be able to benefit more complicated tasks such as QA and reading comprehension:
X. Chen, et. al. WebSRC: A Dataset for Web-Based Structural Reading Comprehension. EMNLP 2021.


**Paper Topic And Main Contributions:**

This paper introduces a new document layout pre-training method that models both text and layout information within document images, thereby enhancing document understanding capabilities. The authors argue the importance of the hierarchical language and layout information of the semi-structured documents. They point out that most previous works rely heavily on the OCR pipelines and propose a novel pre-training method called ViTLP to efficiently and effectively pre-train the model with mixed text-layout sequences. They conduct experiments on multiple VDU tasks, including OCR, information extraction, document classification, and visual question answering.
The main contributions of this paper are as follows:
- Introduction of a VDU pre-training approach that leverages both text and layout information in documents, enhancing document understanding capabilities.
- The pre-trained model that can naturally function as an OCR model, a capability not present in previous works.
- Demonstration of the effectiveness of the proposed method through comprehensive evaluations on OCR tasks and various VDU tasks.
- The multi-segment pre-training scheme enables the model to effectively handle arbitrary long documents.

**Questions For The Authors:**

1. How and why do you use multi-segment fine-tuning in document classification tasks? As for as I am concerned, the decoder of your model will only need to generate the class label in these tasks and the length of the document will not be a problem. The multi-segment approach appears to be exclusively applicable to sequence labeling tasks such as information extraction. Its utility seems limited for tasks involving summarization or free-form generation.
2. In lines 94-97, you claim that your multi-segment pre-training scheme is "more feasible than other long-sequence modeling workarounds such as sparse attention". Why does this claim hold? Could you provide some experiment results to validate this claim?
3. Could you please include the two baselines in your results and give a detailed analysis based on the comparison, or give the reason why not?
4. A few clarifications: Is the 12-layer ViT trained from scratch? What is the input image size in downstream tasks? Are they all scaled to 1600x1280?


**Reasons To Accept:**

1. They propose a novel pre-training method that may better leverage the hierarchical layout information by the mixed prediction of both text and layout (i.e., bounding box).
2. They propose a multi-segment pre-training scheme that can better leverage long documents during pre-training.
3. They propose a two-stage decoding scheme to tackle the so-called token-inefficiency problem when interleaving predicting text tokens and bounding box coordinates.


**Reasons To Reject:**

1. Poor writing. The writing of the paper is not clear in some of the crucial details. For example, the term token-inefficiency appears multiple times in the paper, but do not get well explained. Another example is that they claim to utilize multi-segment fine-tuning (which is not well-defined either, and I infer that is something similar to the multi-segment pre-training scheme they propose) in the document classification tasks. However, they do not show how and why to use it.
2. Relatively poor performance. Although most of their results are the highest performance in the table they present, there are models whose performances are much higher which is not included in these tables. For example, in the tasks presented in Table 2, LayoutLMV3-base (133M, first publicly available in April 2022) has reached 90.29, 96.56, and 95.44, respectively. Higher than the performance of ViTLP with much fewer parameters. As for the VQA tasks (presented in Table 4), Pix2Struct-base (282M, first publicly available in October 2022) has reached 72.1, which is also a generative model.
3. Suspiciously lacking baselines. Usually, not reaching the highest performance will not be a strong reason for rejection. However, the author should include these baselines and analysis the pros and cons, or they should explain why not to include them. However, both of the above are missing in this paper. To make the situation more suspicious, they did cite the paper of both LayoutLMV3 and Pix2Struct (e.g., in line 46, line 56, and more), which shows that they are aware of these two works. That makes me wonder the reason they avoid mentioning these works in their results.
4. Argument without validation. In lines 94-97, the authors claim that their multi-segment pre-training scheme is "more feasible than other long-sequence modeling workarounds such as sparse attention". However, they do not conduct any comparison experiments to validation this claim.
5. The proposed model demonstrates a rigid pattern of generating locations consistently after every word, restricting its ability for free-form generation or transferring to more diverse tasks.

**Reproducibility:**

4: Could mostly reproduce the results, but there may be some variation because of sample variance or minor variations in their interpretation of the protocol or method.

**Reviewer Confidence:**

4: Quite sure. I tried to check the important points carefully. It's unlikely, though conceivable, that I missed something that should affect my ratings.

---

> ### Author Rebuttal · Authors · 2023-08-29
>
> **Rej1: Poor writing**
>
> **Reply:** (1) “Token-inefficiency” means that a document usually contains intensive words (i.e., can be 1K+ BPE subword tokens after tokenization). If we **naively generate four layout-coordinate tokens following each word, the sequence would become especially long**, which quadratically increases the computational and space overhead of the Transformer decoder. We mentioned this point in Lines 148 - 154 and the token-compressed ratio of our approach in Lines 221 - 225. We agree with the reviewer that “token-inefficiency” was not detailly explained and will add detailed explanations to the revised manuscript.
>
> (2) **The multi-segment mechanism is used in the fine-tuning of OCR and information extraction tasks.** We admit that the writing of `multi-segment fine-tuning` is not detailed enough. It should be explained in the following aspects:
>
> - In the approach section, we describe `multi-segment fine-tuning` in Lines 271 - 275. To avoid misunderstanding, the sub-title “Preparing Segment Sequences” of Line 266 should be “Segmentation in Pre-training and Fine-tuning”. We agree that the writing is not good enough and will provide more discussions in the subsection of Lines 271 - 275.
>
> - In the experiment section, we did mention that `multi-segment fine-tuning` can be applied to tasks of form and receipt understanding, i.e., CORD and FUNSD, in Lines 381 - 387 (but not for document classification). **Apparently, the document classification task does not need multi-segmentation.**
>
> - As far as we know, LayoutLM series models truncate the FUNSD and CORD text tokens when exceeding the max sequence length of 512. Please refer to https://github.com/microsoft/unilm/blob/33781165e37f8b9e3514f0aeee642f2c71f1dc18/layoutlmft/examples/run_funsd.py#L169 for details. The multi-segment mechanism enables ViTLP to handle the long documents on the information extraction tasks without truncation. Besides, long document OCR is common, where the multi-segment mechanism is crucial to outputting the whole OCR results without the limitation of Transformer sequence lengths.
>
> We thank the reviewer’s comments on the writing and will add the above discussed details to the revised manuscript.
>
>
>
>
> **Rej2: Relative poor performance and Rej3: Suspiciously lacking baselines**
>
> **Reply:** (1) **On comparison to LayoutLM-v3.** Thank you for the suggestion of comparison with LayoutLM-v3 [1]. For the information extraction task (FUNSD and CORD), we would like to emphasize the important experiment settings of using **(i) entity segment-level layout inputs** and **(ii) word-level layout inputs**. It is worth noting that ViTLP and all compared baselines use word-level rather than segment-level layout inputs. In contrast, LayoutLM-v3 uses segment-level layout inputs.
>
> Note that, there is a considerable performance gap between using entity segment-level and word-level layout inputs. We strongly recommend the reviewer refer to Appendix A.2 of the FormNet-v2 paper [2] and check the performances of these two implementation settings. The performance gap reported in the FormNet-v2 paper is as below.
>
> |Method                 |Setting                             |FUNSD|CORD|
> |---------------------|-----------------------------|---------|-------|
> |LayoutLMv3-base|Reported (segment-level)|90.29    |96.56 |
> |LayoutLMv3-base|Word-level                       |78.35    |85.81 |
> |LayoutLMv3-large|Reported (segment-level)|92.08    |97.46 |
> |LayoutLMv3-large|Word-level                       |82.53    |95.92 |
>
>
> LayoutLM-v3 adopts the segment-level layout position inputs. In Section 2.1 of the LayoutLM-v3 paper [1], the authors explain the reason for this performance gap:
>
> >“The LayoutLM and LayoutLMv2 adopt word-level layout positions, where each word has its positions. Instead, we adopt segment-level layout positions that words in a segment share the same 2D position since the words usually express the same semantic meaning.”
>
> In FUNSD and CORD, multiple entities of the same segment share the same semantic labels. In this setting, using the joint segment-level layout positions can significantly improve entity-level F1 scores. According to the performance of word-level setting reported in the FormNet-v2 paper (refer to Table 1 and Appendix A.2 of the FormNet-v2 paper [2]), the performance of LayoutLM-v3 and ViTLP is compared as below.
>
> |Method                 |Setting                             |FUNSD|CORD|
> |---------------------|-----------------------------|---------|-------|
> |LayoutLMv3-base|Word-level                       |78.35    |85.81 |
> |LayoutLMv3-large|Word-level                       |82.53    |95.92 |
> |ViTLP                   |Word-level                       |86.67    |95.16 |
>
> **Under the same word-level setting,** ViTLP performs the best on FUNSD and LayoutLMv3-large performs the best on CORD. We ASSURE to add performance comparison with LayoutLMv3 in the revised manuscript.
>
>
> (2) **On comparison to Pix2Struct.** Yes, ViTLP underperforms Pix2Struct [3] by a clear margin. Nonetheless, we would like to clarify three perspectives.
>
> - On the pre-training data quantity, Pix2Struct [2] uses 80M images in pre-training, while ViTLP was trained on around 14.3M document images.
>
> - On the pre-training data quality, Pix2Struct uses web-page images with precise annotated texts (because the texts are directly obtained from source HTML), while our pre-trained OCR data naturally contains considerably more noise.
>
> - On the pre-training resources, Pix2Struct-base was trained with a batch size of 2048 on 64 Google Cloud TPUs, while ViTLP was trained with a batch size of 384 on 32 Nvidia-V100 GPUs.
>
> We hope the reviewer can also take the pre-training data and resources into account.
>
>
> (3) **General functionalities and performance of ViTLP.** To the best of our knowledge, ViTLP is the first general document model to perform the tasks of OCR, document understanding (FUNSD, CORD, and RVL-CDIP), and generative document VQA (with explainable ROI outputs). Besides, ViTLP is able to process long documents and generate layout-level outputs. Though ViTLP could not achieve SOTA performance on some tasks (e.g., underperform DONUT as shown in Table 4), we hope the reviewer can take the model’s general functionalities and performance into account.
>
>
>
> **Rej4: Argument without validation**
>
> **Reply:** We agree that the writing is not comprehensive enough. In lines 94 - 97, we intend to discuss the long-document modeling challenge, where sparse attention [4, 5] is a popular workaround. **Though sparse-attention transformers can simplify the self-attention complexity for long-document processing, they need to modify the Transformer architecture and self-attention computation, which may affect the performance of large-scale pre-trained models.** To the best of our knowledge, most (not all) pre-trained models on specific domains (e.g., LayoutLM series [1,6,7] and DONUT [8]) practically follow the standard Transformer architectures (e.g., adapting from the standard Transformer libraries such as HuggingFace, FairSeq, Google t5x, etc.) with task-specific input data and tailored training objectives. Our proposed multi-segment pre-training/fine-tuning mechanism also does not need to modify the Transformer architecture.
>
> We will add detailed discussions on long-document modeling to the revised manuscript. We also hope the reviewer can appreciate that experiments on pre-training are highly expensive. Hence, we could not try Transformer variants for long-document modeling, such as sparse attention Transformers. Training the standard Transformer architecture with multi-segment mechanism is a feasible solution.
>
>
>
> **Rej5: The proposed model demonstrates a rigid pattern of generating locations consistently after every word, restricting its ability for free-form generation or transferring to more diverse tasks.**
>
> **Reply:** (1) In general, some downstream tasks can benefit from the layout information modeling, such as OCR, information extraction, and VQA with reference bounding boxes. While some other tasks, such as free-form generation, do not need the layout modeling ability. As a general approach, ViTLP can also be adapted to the tasks without layout modeling. In these tasks, ViTLP can treat the [LOC] token as a placeholder token and models on text information only.
>
> (2) On the other hand, suppose that layout modeling is not introduced in the pre-training stage, it would be very hard for the model to learn layout modeling ability on fine-tuning tasks. Hence, for the general purpose of text-layout modeling, layout modeling ability in ViTLP is **necessary for pre-training** and **optional for fine-tuning**.
>
>
>
> **Q1: How and why do you use multi-segment fine-tuning in document classification tasks? As for as I am concerned, the decoder of your model will only need to generate the class label in these tasks and the length of the document will not be a problem. The multi-segment approach appears to be exclusively applicable to sequence labeling tasks such as information extraction. Its utility seems limited for tasks involving summarization or free-form generation**
>
> **Reply:** (1) Multi-segment fine-tuning is **NOT** used in the document classification task. It is used in the OCR and information extraction tasks. In both pre-training and fine-tuning, the suffix tokens of the previous segments serve as prefix tokens for the subsequent token generation (or inference). For details, please refer to our response point (2) to “Rej1”.
>
> (2) ViTLP can also be adapted to the tasks without layout modeling. Layout modeling of ViTLP is necessary for pre-training and optional for fine-tuning. For details, please refer to our response to “Rej5”.
>
>
>
> **Q2: In lines 94-97, you claim that your multi-segment pre-training scheme is "more feasible than other long-sequence modeling workarounds such as sparse attention". Why does this claim hold? Could you provide some experiment results to validate this claim?**
>
> **Reply:** Please refer to our detailed response to “Rej4”.
>
>
>
> **Q3: Could you please include the two baselines in your results and give a detailed analysis based on the comparison, or give the reason why not?**
>
> **Reply:** Please refer to our detailed response to “Rej2&3”.
>
>
>
> **Q4: A few clarifications: Is the 12-layer ViT trained from scratch? What is the input image size in downstream tasks? Are they all scaled to 1600x1280?**
>
> **Reply:** (1) We use pre-trained ViT/32-base from https://huggingface.co/google/vit-base-patch32-384 to initialize the ViTLP image encoder. Thanks for this question, and we will add this detail to the revised manuscript. In our pilot experiments, we found that using (or not using) pre-trained ViT did not significantly affect the final training loss (in 100K steps), but using pre-trained ViT did help the training loss descend and converge faster.
>
> (2) In both pre-training and fine-tuning, the input image resolutions are scaled to 1600x1280.
>
>
>
>
> [1] LayoutLMv3: Pre-training for Document AI with Unified Text and Image Masking, ACM Multimedia 2022.
>
> [2] FormNetV2: Multimodal Graph Contrastive Learning for Form Document Information Extraction, ACL 2023.
>
> [3] Pix2Struct: Screenshot Parsing as Pretraining for Visual Language Understanding, ICLM 2023.
>
> [4] Generating Long Sequences with Sparse Transformers, OpenAI 2019.
>
> [5] Longformer: The Long-Document Transformer, Arxiv 2020.
>
> [6] LayoutLM: Pre-training of Text and Layout for Document Image Understanding, KDD, 2020
>
> [7] LayoutLMv2: Multi-modal Pre-training for Visually-Rich Document Understanding, ACL, 2021
>
> [8] OCR-free Document Understanding Transformer, ECCV 2022.

---

### Official Review · Reviewer_a82W · 2023-08-10

**Soundness:** 2

**Ethical Concerns:**

Yes

**Excitement:**

3: Ambivalent: It has merits (e.g., it reports state-of-the-art results, the idea is nice), but there are key weaknesses (e.g., it describes incremental work), and it can significantly benefit from another round of revision. However, I won't object to accepting it if my co-reviewers champion it.

**Paper Topic And Main Contributions:**

The paper introduces a unified OCR-free approach called visually guided generative text-layout pre-training (ViTLP), which aims to jointly model text and layout information in document images. By optimizing hierarchical language and layout modeling objectives, ViTLP generates a combined target sequence of texts and layouts. Additionally, it can function as an OCR tool for text detection and recognition in document images. Furthermore, the proposed approach effectively handles documents of varying lengths through multi-segment generative pre-training. Experimental results show that ViTLP outperforms baselines on a range of visual document processing tasks.

**Questions For The Authors:**

You have used a privatized OCR tool to extract texts and layouts, which is the key supervision for pretraining. Will you release such pre-processed data to ensure the reproducibility of your paper?

**Reasons To Accept:**

 1. It introduces a novel approach for visual document processing, wherein mixed text-layout sequences are generated from document images using an encoder-decoder architecture. The encoder is responsible for extracting image features, while the decoder, equipped with different prefixes/prompts, can handle pre-training and various downstream tasks within a unified model.
2. The hierarchical text-layout decoder employed in this study is intriguing. The global text-layout modeling module generates mixed target text-layout sequences, while the local layout modeling module focuses on generating specific layout coordinates for each layout token.

**Reasons To Reject:**

1.The performance improvements are not strong  and the experimental setup is not convincing.  This paper does not compare with the top-performing methods, such as LayoutLM v3 and FormNet v2, where the F1 score of LayoutLM v3-base on the FUNSD and CORD datasets are 90.29% and 96.56%, respectively.
2.It unclear whether the impressive performance is largely due to the extra high-quality datasets introduced, such as IAM and DocBank. For fair comparisons, it is recommended to report the results that specifically showcase the performance of the proposed method when solely relying on the most commonly-used IIT-CDIP Test Collection 1.0 for pre-training. This would allow for a more comprehensive evaluation of the method's effectiveness.



**Reproducibility:**

3: Could reproduce the results with some difficulty. The settings of parameters are underspecified or subjectively determined; the training/evaluation data are not widely available.

**Reviewer Confidence:**

3: Pretty sure, but there's a chance I missed something. Although I have a good feel for this area in general, I did not carefully check the paper's details, e.g., the math, experimental design, or novelty.

---

> ### Author Rebuttal · Authors · 2023-08-29
>
> **Q1: Regarding the experiment performance.**
>
> **Reply:** Thanks for your constructive suggestion of comparison with LayoutLM-v3 [1] and FormNet-v2 [2] on FUNSD and CORD. Preliminarily, we would like to emphasize the important experiment settings of using **(i) entity segment-level layout inputs** and **(ii) word-level layout inputs**. It is worth noting that ViTLP and all compared baselines use word-level rather than segment-level layout inputs. In contrast, LayoutLM-v3 and FormNet-v2 use segment-level layout inputs.
>
> Note that, there is a considerable performance gap between using entity segment-level and word-level layout inputs. We strongly recommend the reviewer refer to Appendix A.2 of the FormNet-v2 paper [2] and check the performances of these two implementation settings. The performance gap reported in the FormNet-v2 paper is as below.
>
> |Method                 |Setting                             |FUNSD|CORD|
> |---------------------|-----------------------------|---------|-------|
> |LayoutLMv3-base|Reported (segment-level)|90.29    |96.56 |
> |LayoutLMv3-base|Word-level                       |78.35    |85.81 |
> |LayoutLMv3-large|Reported (segment-level)|92.08    |97.46 |
> |LayoutLMv3-large|Word-level                       |82.53    |95.92 |
>
>
> LayoutLM-v3 adopts the segment-level layout position inputs. In Section 2.1 of the LayoutLM-v3 paper [1], the authors explain the reason for this performance gap:
>
> >“The LayoutLM and LayoutLMv2 adopt word-level layout positions, where each word has its positions. Instead, we adopt segment-level layout positions that words in a segment share the same 2D position since the words usually express the same semantic meaning.”
>
> In FUNSD and CORD, multiple entities of the same segment share the same semantic labels. In this setting, using the joint segment-level layout positions can significantly improve entity-level F1 scores. According to the performance of word-level setting reported in the FormNet-v2 paper (refer to Table 1 and Appendix A.2 of the FormNet-v2 paper [2]), the performance of LayoutLM-v3, FormNet-v2, and ViTLP is compared as below.
>
> |Method                 |Setting                             |FUNSD|CORD|
> |---------------------|-----------------------------|---------|-------|
> |LayoutLMv3-base|Word-level                       |78.35    |85.81 |
> |LayoutLMv3-large|Word-level                       |82.53    |95.92 |
> |FormNetV2          |Word-level                       |86.35    |**97.37** |
> |ViTLP                   |Word-level                       |**86.67**    |95.16 |
>
> Under the same word-level setting, ViTLP performs the best on FUNSD and FormNetV2 performs the best on CORD.
>
>
> **Q2: It is unclear whether the impressive performance is largely due to the extra high-quality datasets introduced, such as IAM and DocBank.**
>
> **Reply:** Thanks for this valuable question. First, we would like to list the dataset statistics
> |Dataset      |# Doc Images|
> |-------------|----------------|
> |IIT-CDIP     |11.5M           |
> |SynthoDog|2M                 |
> |DocBank   |0.2M              |
> |IAM            |1.2K              |
> |SciTSR      |3.5K              |
> |PublayNet |0.5M             |
>
> ViTLP uses the same amount of IIT-CDIP (11.5M) and SynthDog (2M) data as DONUT [3]. The IIT-CDIP and SynthoDog dataset sizes are mentioned in Lines 330 and 341, respectively. These two datasets account (13.5M) for the largest proportions compared to all other datasets (0.8M). Hence, the overall dataset usage should be fair. Since the pre-training costs are very expensive, we could not pre-train another model only based on IIT-CDIP (and SynthDog) to strictly follow the dataset setting of DONUT. We hope the reviewer can appreciate this.
>
> Besides, we will include the dataset statistics in an appendix section in the revised manuscript.
>
>
>
> **Q3: Release of pre-processed OCR data.**
>
> **Reply:** Yes, we ASSURE to open-source pre-process data and scripts for reproducibility.
>
>
>
>
> [1] LayoutLMv3: Pre-training for Document AI with Unified Text and Image Masking, ACM Multimedia 2022.
>
> [2] FormNetV2: Multimodal Graph Contrastive Learning for Form Document Information Extraction, ACL 2023.
>
> [3] OCR-free Document Understanding Transformer, ECCV 2022.

---

### Official Review · Reviewer_Ks4h · 2023-08-11

**Soundness:** 2

**Excitement:**

3: Ambivalent: It has merits (e.g., it reports state-of-the-art results, the idea is nice), but there are key weaknesses (e.g., it describes incremental work), and it can significantly benefit from another round of revision. However, I won't object to accepting it if my co-reviewers champion it.

**Missing References:**

LayoutTransformer: Layout Generation and Completion with Self-attention. This paper adopts a similar approach for generating tokens with positions and is not included in the related work.


**Paper Topic And Main Contributions:**

This paper proposes a new modeling approach called ViLTP for visual document understanding.
- It employs an encoder-decoder architecture that can encode both the document image, text, layout and can generate text-layout sequences (i.e., generating the token and the position of the token simultaneously)
- It uses a hierarchical language and layout modeling objective to efficiently generate the token position information. Besides the main transformer decoder, it utilizes a separate RNN model to decode the position information (x,y coordinates of the token) from a position token [POS] generated by the main transformer model.
- It has an additional multi-segment Pre-training schema to help the model use text from overlapping sliding windows when truncating the input with fixed windows.

Empirical results show that this approach can achieve similar or better performance compared to other baseline methods on different tasks like OCR, VDU, and DocVQA.


**Questions For The Authors:**

1. It is not appropriate to claim the ViTLP model has unlimited doc-length (table 2). The multi-segment Pre-training schema in ViTLP uses overlapping sliding windows when truncating text, and the maximum tokens it can take at one time is only 1024 (line 320).
2. What is the output vocabulary size of the RNN model? If it is generating numerical coordinates, are there any benefits of making it a classification task rather than a regression task? (line 209)
3. It would be great if the authors can provide additional details on the training of the RNN models. Is it trained jointly? (line 313)
4. The examples in Figure 3 might contain errors.
    1. For the middle figure, is there a typo for the 4th coordinate (447) in the ViLTP output? If not, then the shape of the red rectangle drawn might not be correct.
    2. Is the image in the figure 3 cropped? If not, I find the coordinates generated by the model might not be correct. According to the figure, (line 332), the location coordinates are normalized to [0, 1000]. Comparing the left and middle figure, it seems the x coordinates of “19,000” in the first figure should be around 400 instead of 200 in the illustration.
    3. I’d recommend the authors provide additional examples (including failure cases) of the model.
5. It would be great if the authors can share examples for OCR results of the model. (table 1).
6. (line 443) The authors claim “This result indicates that ViTLP can develop a thorough understanding of form structures in document images.” Is there proof other than the slightly better accuracy in the FUNSD dataset?


**Reasons To Accept:**

The proposed ViLTP approach has the following strengths:
- It fuses image, text, and layout that leads to strong performance compared to existing methods.
- The versatility of the method can be helpful for many downstream tasks.
- Text generation with layout information can provide helpful evidence for referencing the outputs in the source (e.g., figure 3).


**Reasons To Reject:**

- From the modeling perspective, training a transformer model and an additional RNN module jointly can be difficult.
- Requiring generating a [POS] after each real token seems to be redundant and might not be needed for all the tasks. It could be potentially improved by slightly changing the modeling objectives.
- Some claims in the paper are not accurate or properly supported, and I am not convinced by the examples in the provided current draft. Please refer to the questions for the authors below, and I am happy to update my judgment with additional evidence provided.


**Reproducibility:**

3: Could reproduce the results with some difficulty. The settings of parameters are underspecified or subjectively determined; the training/evaluation data are not widely available.

**Reviewer Confidence:**

4: Quite sure. I tried to check the important points carefully. It's unlikely, though conceivable, that I missed something that should affect my ratings.

**Typos Grammar Style And Presentation Improvements:**

- In table 2, ViTLP’s accuracy in RVL-CDIP is not the top but the text is bolded (with DONUT being the top performing model).

---

> ### Author Rebuttal · Authors · 2023-08-28
>
> **Rej1: “Training a transformer model and an additional RNN module jointly can be difficult”.**
>
> **Reply:** Thanks for your valuable question, which we think is exactly what some readers may be concerned about. We address this question from the following aspects:
>
> (1) **The layout-head is a lightweight RNN and works similarly to the LM-head.** In pre-trained language models, Transformers parameterizes task knowledge in deep self-attention and FFN layers, whereas the LM-head functions as a linear classifier to predict the next word. The LM-head itself is a simple linear layer. This also applies to the RNN layout-head, which has only three linear layers (two RNN matrices and a coordinate-prediction matrix, refer to https://pytorch.org/docs/stable/generated/torch.nn.RNNCell.html) to predict the next coordinate. Given the knowledge learned by deep Transformer layers, a simple linear LM-head and RNN layout-head is feasible to predict the next word/coordinate tokens based on learned multimodal representations $\mathbf{H}_{n}^{VLT}$.
>
> (2) **In order to “capture the spatial dependencies among coordinates” as stated in Line 203, we use an RNN head rather than four independent linear heads.** Since the coordinate sequences are especially short (i.e., having only four coordinate tokens), a simple RNN head rather than complex self-attention layers is capable of such short sequence generation.
>
> (3) **From our experimental findings, adding the RNN-head generation loss did not significantly affect the language modeling (LM) loss.** We conducted the ablation experiments of `(i) LM-loss only` (“w/o layout modeling” in Table 3) and `(ii) LM-loss + Layout-loss` as in Eq. (5). The two LM-loss curves of (i) and (ii) descended and converged very closely. We believe that the layout information is learned and preserved in the [LOC] token representations, and hence it does not significantly affect the language modeling loss.
>
> Since this question is intriguing, we will add the above discussion to an appendix section.
>
>
>
> **Rej2: “Requiring generating a [POS] after each real token seems to be redundant and might not be needed for all the tasks”.**
>
> **Reply:** The layout token [LOC] is necessary. In our pilot experiments, we have tried to directly generate a word’s location coordinates from the representations of the word’s last BPE token without [LOC]. Empirically, we found this approach would introduce two serious problems:
>
> (1) **Without the [LOC] token, the model cannot predict the next coordinate token based on the generated tokens, which leads to poor OCR performance.** Note that the model cannot know the last BPE token of a word in the generation stage. For example, with a RoBERTa (or BART) BPE tokenizer, the word “pre-process” is tokenized into BPE tokens of [5234, 12, 31931], while the word “pre-processed” is tokenized into [5234, 12, 31931, 196]. When encountering the token [31931], the model cannot know whether it is the last BPE token of the word. In this situation, we cannot feed the generated coordinate tokens to the model on-the-fly for the next-step generation like ViTLP (in Figure 2). This further makes the Transformer decoder cannot predict the next coordinates based on the previously generated coordinates, which is crucial for layout generation. As we found in the pilot experiments, this issue caused the model’s OCR performance marginally worse than ViTLP with [LOC] tokens.
>
> (2) **Effect of [LOC] on pre-training and fine-tuning.** In our early experimental trials, we compared the pre-training losses of `(i) ViTLP without [LOC]` and `(ii) ViTLP with [LOC]`. The pilot experiments were conducted on a small subset of IIT-CDIP data. We found that if we did not insert [LOC] tokens, the text modeling and layout modeling losses were significantly higher than `ViTLP with [LOC]`. We also tested the pre-trained checkpoints and observed worse performance of the `ViTLP without [LOC]` on FUNSD and CORD. We inferred that it was hard to learn multimodal representation $\mathbf{H}_{n}^{VLT}\in\mathbb{R}^{768}$ simultaneously encoding the text and layout information in one hidden state. Modeling the layout information in a separate token [LOC] could encourage disentangled text and layout generation, which brings lower pre-training loss and better fine-tuning performance.
>
> For comprehensive practice on the Transformer-based mixed token generation, we refer the reviewer to Pix2seq [1] (which we have cited in Line 146).
>
>
>
> **Rej3 and Q4: Regarding the example in Figure 3.**
>
> **Reply:** We appreciate the reviewer carefully reading our paper and noticing this. Yes, the shown image is cropped. We have to crop the images to put all three examples in a row. If not cropped, the images would be too large to make the words too small to identify in Figure 3. As OpenReview does not support image upload, we refer the reviewer to download and check the three original images from https://rrc.cvc.uab.es/?ch=17&com=downloads. The three images are from the DocVQA validation set, we refer the reviewer to check the original files at
>
> `val/documents/jjvg0227_8.png`
>
> `val/documents/lycj0037_6.png`
>
> `val/documents/fzyh0227_7.png`
>
> For the middle image, the 4th coordinate (447) is a typo, it is 747 instead. Many thanks to the reviewer for pointing it out. We will correct it in the revised manuscript.
>
>
>
> **Q1: It is not appropriate to claim the ViTLP model has unlimited doc-length (table 2).**
>
> **Reply** We denote the “Maximum Doc-length” as the maximum tokens that a VDU model can process. Note that baseline models truncate long documents and can process certain numbers of tokens at most (e.g., 512 tokens for LayoutLM-v2). In contrast, as mentioned in Lines 271-275, ViTLP can also process documents with arbitrary lengths by segmentation in both pre-training and fine-tuning.
>
>
>
> **Q2: What is the output vocabulary size of the RNN model? If it is generating numerical coordinates, are there any benefits of making it a classification task rather than a regression task? (line 209)**
>
> **Reply:** As mentioned in Line 323, the vocabulary size of RNN layout-head is 1000. We formulate it as a task of coordinate token generation. We made it a classification task for two reasons:
>
> (1) Token-based autoregressive generation is practical in NLP sequence modeling (e.g., in T5 and GPT). Note that the text modeling head and layout modeling head share a multimodal Transformer decoder. To keep the **layout modeling objective (in Eq. (4))** in line with the **text modeling objective (in Eq. (3))**, we also formulate layout modeling as an autoregressive generation in the same way. As stated in Lines 140 - 144, the joint text and layout modeling can be viewed as a unified autoregressive generation process.
>
> (2) To the best of our knowledge, incorporating location representations by an additional coordinate vocabulary has been a prevalent way in multimodal learning (e.g., OFA [2] and VisionLLM [3]). Note that the generated layout coordinates of the i-th step are fed to the model for the (i+1)-th step generation (refer to the location embedding in Lines 179 - 184 and Figure 2). Pre-trained language models naturally follow this paradigm of next-token prediction based on previously generated tokens, whereas outputting continuous regression values would be incompatible.
>
>
>
> **Q3: It would be great if the authors can provide additional details on the training of the RNN models. Is it trained jointly? (line 313)**
>
> **Reply:** As stated in Eq. (2) and Eq. (5), the text modeling objective and layout modeling objective are summed and trained jointly. For additional details on the RNN layout-head design, please refer to the above response to “Rej1”.
>
>
>
> **Q4: Regarding the examples in Figure 3**
>
> **Reply:** Please refer to the above response to “Rej3”. We agree with the suggestion that providing failure examples would be better to show comprehensive model performance. We ASSURE combining success and failure examples in the revised manuscript.
>
>
>
> **Q5: It would be great if the authors can share examples for OCR results of the model. (table 1).**
>
> **Reply:** We are very sorry that images cannot be provided in OpenReview. We ASSURE providing qualitative OCR examples in an appendix section.
>
>
>
> **Q6: (line 443) The authors claim “This result indicates that ViTLP can develop a thorough understanding of form structures in document images.” Is there proof other than the slightly better accuracy in the FUNSD dataset?**
>
> **Reply:** Thanks for this in-depth question. We address this question from the following perspectives.
>
> (1) As has been proved by prior research (e.g., LayoutLM-v2 [4]), layout information modeling is essential to multimodal document understanding tasks, including information extraction. This is because real-world documents are usually structural: documents typically contain multiple structural elements of headers, content, figures, tables, etc. Different from prior generative document models that only exploit text modeling (e.g., DONUT [5]), ViTLP establishes a thorough understanding of multimodal documents by unifying text and layout modeling in a generative pre-training framework.
>
> (2) As discussed in Lines 226 - 229, the interleaved sequences of mixed text and layout tokens enable compact interaction between text and layout information, which benefits the downstream document understanding tasks. This **interleaved sequence modeling of mixed modality tokens** facilitates effective interaction of multimodal inputs, which can be referred to [6].
>
> (3) As a concrete example mentioned in Lines 450 - 456, layout information is useful for predicting the semantic labels in the information extraction tasks (FUNSD and CORD). Here we provide another concrete example of FUNSD. The header entity of a document is always located in the top center of the document. ViTLP can leverage the corresponding layout coordinate information to better perform entity-level classification on the FUNSD `<header>` semantic label.
>
>
>
>
> [1] Pix2seq: A Language Modeling Framework for Object Detection, ICLR 2022.
>
> [2] OFA: Unifying Architectures, Tasks, and Modalities Through a Simple Sequence-to-Sequence Learning Framework, ICML 2022.
>
> [3] VisionLLM: Large Language Model is also an Open-Ended Decoder for Vision-Centric Tasks, Arxiv 2023.
>
> [4] LayoutLMv2: Multi-modal Pre-training for Visually-rich Document Understanding, ACL 2021.
>
> [5] OCR-free Document Understanding Transformer, ECCV 2022.
>
> [6] Flamingo: a Visual Language Model for Few-Shot Learning, NeurIPS 2022.

---

### Meta-Review · Area_Chair_3v8U · 2023-09-18

**Recommendation:** 1

**Metareview:**

The authors introduce ViLTP, an innovative OCR-free vision-language pretraining objective. This model seeks to cohesively integrate text and layout information from document images, aiming to bolster performance in downstream tasks like document classification and extraction. All reviewers concur that, despite its novel pretraining objective, ViLTP exhibits only marginal empirical performance, particularly in QA tasks. Moreover, the analysis lacks crucial baselines, notably other contemporary foundational models for document comprehension.

While the authors have presented a comprehensive rebuttal, complemented by additional experiments, I am of the opinion that it doesn't fully address the concerns raised by the reviewers.

---

### Decision · Program_Chairs · 2023-10-07

**Decision:**

Reject

**Comment:**

The authors introduce ViLTP, an innovative OCR-free vision-language pretraining objective. This model seeks to cohesively integrate text and layout information from document images, aiming to bolster performance in downstream tasks like document classification and extraction. All reviewers concur that, despite its novel pretraining objective, ViLTP exhibits only marginal empirical performance, particularly in QA tasks. Moreover, the analysis lacks crucial baselines, notably other contemporary foundational models for document comprehension.

While the authors have presented a comprehensive rebuttal, complemented by additional experiments, I am of the opinion that it doesn't fully address the concerns raised by the reviewers.